# Anatomy and activity patterns in a multifunctional motor neuron and its surrounding circuits

**Mária Ashaber[1†], Yusuke Tomina[1†‡], Pegah Kassraian[1†], Eric A Bushong[2], William B Kristan[2], Mark H Ellisman[3,4], Daniel A Wagenaar[1]\***

[1]Division of Biology and Biological Engineering, California Institute of Technology, Pasadena, United States; [2]Division of Biological Sciences, University of California, San Diego, San Diego, United States; [3]National Center for Microscopy and Imaging Research, University of California, San Diego, San Diego, United States; [4]Department of Neurosciences, UCSD School of Medicine, San Diego, United States

**Abstract** Dorsal Excitor motor neuron DE-3 in the medicinal leech plays three very different dynamical roles in three different behaviors. Without rewiring its anatomical connectivity, how can a motor neuron dynamically switch roles to play appropriate roles in various behaviors? We previously used voltage-sensitive dye imaging to record from DE-3 and most other neurons in the leech segmental ganglion during (fictive) swimming, crawling, and local-bend escape (Tomina and Wagenaar, 2017). Here, we repeated that experiment, then re-imaged the same ganglion using serial blockface electron microscopy and traced DE-3's processes. Further, we traced back the processes of DE-3's presynaptic partners to their respective somata. This allowed us to analyze the relationship between circuit anatomy and the activity patterns it sustains. We found that input synapses important for all the behaviors were widely distributed over DE-3's branches, yet that functional clusters were different during (fictive) swimming vs. crawling.

**\*For correspondence:**
daw@caltech.edu

[†]These authors contributed equally to this work

**Present address:** [‡]Faculty of Science and Technology, Keio University, Yokohama, Japan

**Competing interests:** The authors declare that no competing interests exist.

## Introduction

Some neural circuits are responsible for only one specialized function. Examples include the exquisitely tuned delay lines that barn owls use to locate sounds (*Carr and Konishi, 1988*) and the motor neurons that control the honeybee's stinger (*Ogawa et al., 1995*). In the early days of neuroscience, those circuits received most attention, likely because they are in important ways more tractable. In most animals' central nervous systems, however, many circuits respond to stimuli of multiple sensory modalities or control more than one behavior (*Briggman and Kristan, 2008*). Increasingly, even sensory brain areas once considered unimodal are found to be modulated by or to directly respond to other sensory modalities, or to be modulated by behavioral state. For instance, auditory stimuli can modulate activity in the human visual cortex (*Plass et al., 2019*) (in sighted as well as in blind subjects [*Bedny et al., 2015*]). Likewise, motor cortex activity can be modulated not just by visual presentation of images of relevant body parts, but even in a working memory task in the absence of immediate stimuli (*Galvez-Pol et al., 2018*). All these forms of multifunctionality likely contribute to the versatility of large brains like ours and may help smaller animals use their more constrained neural hardware more efficiently (*Briggman and Kristan, 2008*).

A particularly interesting situation occurs when multifunctional circuits converge onto a single motor neuron which plays distinct roles in different behaviors (*Miroschnikow et al., 2018*; *Zarin et al., 2019*). What functional and anatomical aspects of the converging pathways permit the output neuron to play its various roles? Control of locomotion in the medicinal leech (*Hirudo verbana*) is a prime example of a system organized in this manner. In each of the animal's segmental

ganglia, partially overlapping circuits generate the rhythms for swimming and crawling as well as the dynamics of an escape behavior known as the local bend (*Kristan et al., 2005*; *Briggman et al., 2005*; *Tomina and Wagenaar, 2017*). These circuits have a common output in a motor neuron called DE-3 that excites the animal's dorsal longitudinal muscles. DE-3 plays markedly different dynamical roles in each of the behaviors: In swimming, it oscillates at a relatively rapid 2 Hz in antiphase to its Ventral Excitor counterpart (VE-4) and controls the local dorsal flexion phase of the swim rhythm. In crawling, it is active in phase with VE-4 and controls the contraction phase of the crawl rhythm. In 'local bending' it operates in tight concert with its contralateral homolog as well as VE-4 to precisely deform the local body wall away from a mechanical stimulus.

To study the mechanisms underlying the versatility of multifunctional circuits, one would ideally like to record from every single neuron in a nervous system during all of the behaviors the animal can execute, and then reconstruct the anatomical connections between those neurons. Once a mere dream, this is rapidly becoming practicable: Activity imaging using calcium dyes has advanced to the point where simultaneous recordings from the vast majority of individual neurons in smaller species can be accomplished, for instance in larval zebrafish (*Ahrens et al., 2013*). This technique has even been applied to behaving animals (*Jiao et al., 2018*). Likewise, anatomical imaging using electron microscopy (EM) has advanced to the point that brains as large as that of the fruit fly *Drosophila melanogaster* can be imaged—and substantial fractions of their circuitry reconstructed—at a synaptic resolution (*Zheng et al., 2018*; *Bates et al., 2020*; *Scheffer et al., 2020*; *Maniates-Selvin et al., 2020*). Even small vertebrate brains like that of larval zebrafish are yielding to this approach (*Hildebrand et al., 2017*).

To bring form and function together, an anatomical atlas based on electron microscopy can be combined with functional studies that use the split-GAL4 system to target individual neurons and identify them with neurons in the anatomical map (*Eichler et al., 2017*; *Eschbach et al., 2020*). This is an immensely powerful approach because a single (costly and time consuming) EM run can be used with an unlimited number of functional studies. However, cross-identification of neurons between EM and functional imaging in this approach is ultimately limited by the fact that all nervous systems, even the simplest ones, exhibit considerable variability in their connectomes (*Bhattacharya et al., 2019*). Accordingly, robustly linking function to connectivity requires a combined anatomical and functional assessment within the same animal (*Bargmann and Marder, 2013*). Nearly a decade ago, back-to-back papers in *Nature* described two-photon calcium imaging in the mouse visual cortex followed by serial transmission electron microscopy of a small volume of that same cortex (*Bock et al., 2011*) and two-photon calcium imaging in the mouse retina followed by serial block face electron microscopy of that same retina (*Briggman et al., 2011*). This approach, 'Correlated Light and Electron Microscopy' or CLEM (*de Boer et al., 2015*), has since allowed the quantification of synaptic connections between functionally identified neurons that control eye movement in the larval zebrafish (*Vishwanathan et al., 2017*), a reconstruction of the connections in the olfactory bulb of another larval zebrafish in which neural responses to eight different odors had been recorded prior to sectioning (*Wanner and Friedrich, 2020*), and a critical re-assessment of the mechanisms of selectivity to particular sensory features in neurons of the mammalian visual cortex (*Scholl et al., 2019*).

For the present study of multifunctional behavioral circuits, we chose to use a voltage-sensitive dye (VSD) (*Grinvald and Hildesheim, 2004*; *Kulkarni and Miller, 2017*) rather than a calcium dye because it can capture faster temporal dynamics, potentially even recording individual action potentials, as well as both excitatory and inhibitory synaptic potentials. We used a fast, high-sensitivity VSD (*Woodford et al., 2015*) to record the neuronal activity in a segmental ganglion of the medicinal leech *Hirudo verbana* while its nervous system expressed several (fictive) behaviors. Then, to map the circuits underlying those behaviors with synapse-level resolution, we re-imaged the same individual ganglion using serial blockface electron microscopy (SBEM) (*Lippens et al., 2019*). Finally, we traced the full arborization of the DE-3 motor neuron and the relevant parts of all its presynaptic partners, to enable comparison between functional and anatomical features of the circuit. *Figure 1* illustrates the overall approach. To our knowledge, this is the first time VSDs and SBEM have been combined at this scale.

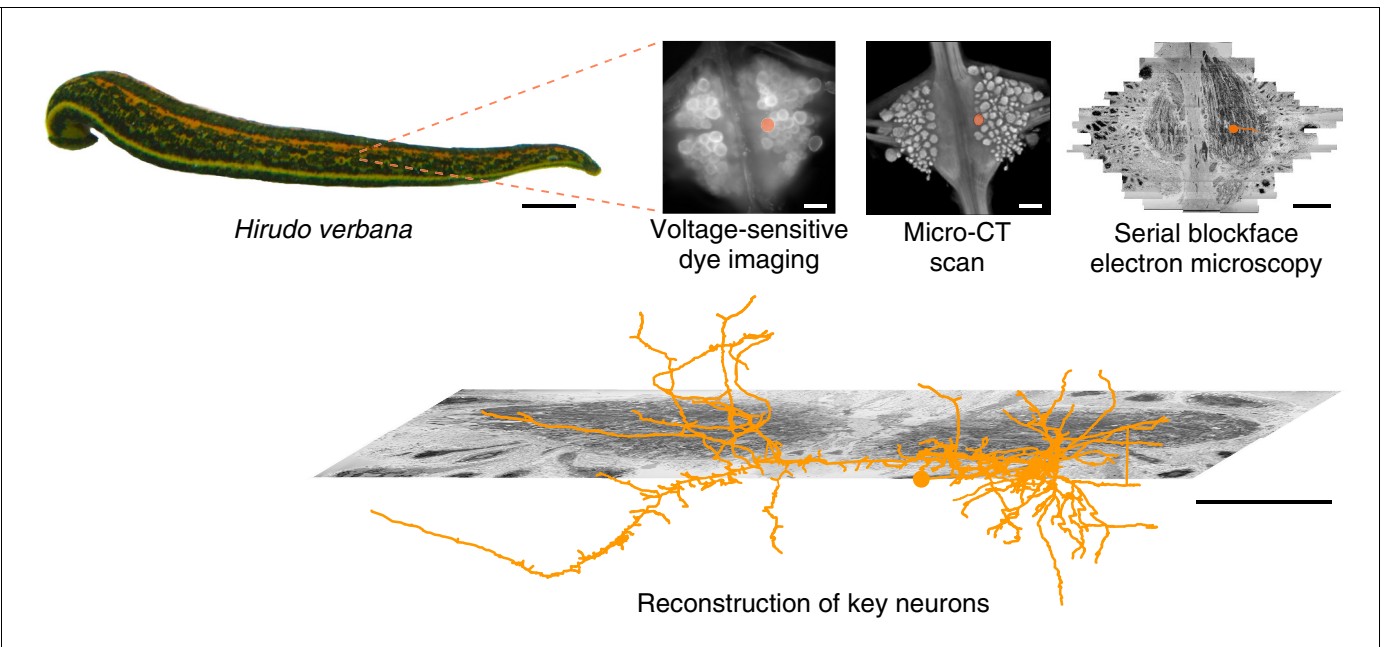

**Figure 1.** Approach. Several fictive behaviors were induced in the isolated nervous system of a medicinal leech while one segmental ganglion was imaged using a voltage-sensitive dye. After fixation and resin embedding, the ganglion was X-ray-imaged to verify that the geometry of somata was preserved. Finally, the neuropil was imaged at nanometer resolution with serial blockface electron microscopy and selected neurons were manually traced. Scale bars: 1 cm (leech photograph); 100 μm (all others).

## Results

### Voltage-dye imaging of behavior

The nervous system of the leech comprises cephalic ganglia, a tail ganglion, and 21 nearly identical segmental ganglia connected by a ventral nerve cord (*Muller et al., 1981*; *Wagenaar, 2015*). Each ganglion consists of about 400 neurons (*Macagno, 1980*) with cell bodies arranged in a spherical monolayer around a central neuropil. In the neuropil, neurons communicate through chemical and electrical synapses located along extensively branched neurites (*Muller and McMahan, 1976*; *Fan et al., 2005*; *Pipkin et al., 2016*). The leech is an ideal model organism for this type of work, because it robustly expresses several behaviors even in reduced preparations (*Kristan et al., 2005*), its neurons are uncommonly accessible to physiological recording, and its cell bodies are relatively large and thus yield strong VSD signals (*Briggman et al., 2005*). Crucially, an individual segmental ganglion is a good stand-in for a whole nervous system, because its neurons capture the entire pathway from sensory input through self-generated interneuronal rhythms to motor output (*Kristan et al., 2005*), which is why we focused our imaging efforts there.

We expressed (fictive) swimming, crawling, and local bending behavior in the isolated nervous system of a single adult leech following the same protocol used for a previous extensive study of these behaviors using VSD imaging in a larger group of animals (*Tomina and Wagenaar, 2017*). As in the previous study, one segmental ganglion in the chain was prepared for VSD imaging and we recorded from both the ventral and dorsal aspects simultaneously with a double-sided fluorescence microscope (*Figure 2a*). We were able to record from 250 neurons simultaneously, similar to our previous results. Fictive swimming was induced by electrical stimulation of a posterior segment, which resulted in characteristic rhythmic activity in dorsal motor neurons and many other neurons on both sides of the ganglion (*Figure 2b*). Coherence analysis confirmed that the rhythms of the various neurons were indeed related to the fictive behavior (*Figure 2c,d*). In a similar manner, we induced fictive crawling (*Figure 3*) and local bending (*Figure 4*).

We established a mapping between the neurons seen in the VSD images and the canonical maps of the ganglion (*Wagenaar, 2017*) based on geometry and on the involvement of the neurons in the various behaviors.

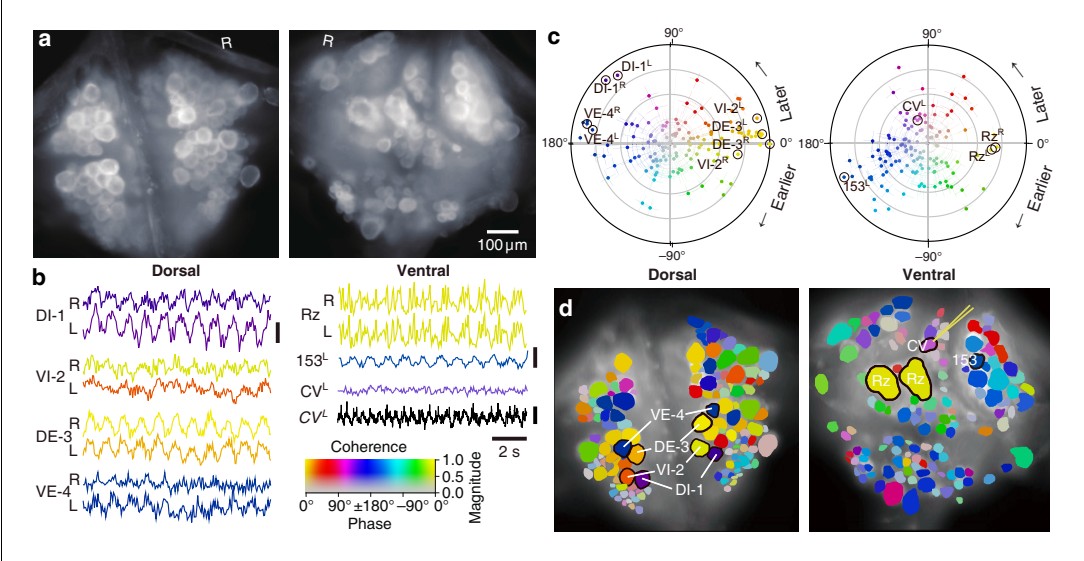

**Figure 2.** Fictive swimming imaged using VSD. (a) Images of the dorsal (*left*) and ventral (*right*) aspects of a leech ganglion simultaneously obtained using a double-sided microscope. 'R' indicates the right side of the ganglion (i.e., the animal's right when dorsal side up). (b) Selected VSD traces during fictive swimming. From the dorsal surface: dorsal and ventral inhibitory and excitatory motor neurons DI-1, VI-2, DE-3, and VE-4; from the ventral surface: the Retzius cells (neuromodulatory interneurons) and cell 153$^L$ (an interneuron). All those cells are known to be rhythmically active during swimming. Also shown is CV$^L$, an excitor of ventrolateral circular muscles that was intracellularly recorded during the trial as a control to verify that fluorescence signals reflect membrane potential changes as they should. Scale bars: 0.2% relative fluorescence change, 5 mV membrane potential change. (c) Magnitude (radial axis from 0 to 1) and phase (angular coordinate) of the coherence of activity in individual neurons with the swim rhythm in motor neuron DE-3$^R$. Error bars indicate confidence intervals based on a multi-taper estimate. (d) Coherence maps of the VSD signals of all cells on the dorsal (*left*) and ventral (*right*) surfaces of the ganglion. Colors of cell bodies indicate coherence relative to DE-3$^R$. The yellow needle on CV$^L$ indicates a sharp electrode for intracellular recording. Color scale applies to all panels.

## X-ray tomography connects functional and anatomical image stacks

At the end of the (fictive) behavior experiment, the ganglion was fixated and embedded in a resin. To correlate light and electron microscopy, we then re-imaged the ganglion using X-ray tomography (*Bushong et al., 2015*) and verified that the cell bodies seen in the VSD images could still be identified (*Figure 5a*). The X-ray image stack was also used to trace neuronal processes from the somata to the edge of the neuropil (*Figure 5b*). This obviated the need to capture the somata in the subsequent electron microscopy, and instead allowed us to restrict the EM effort largely to the neuropil.

## Electron microscopy

We chose serial blockface electron microscopy (SBEM) over serial-section transmission EM (*Harris et al., 2006*) because SBEM can reliably process large numbers of slices with much lower risk of sectioning artifacts. We acquired 78,803 images from 9604 slices, totaling 22.8 terapixels. We periodically paused the acquisition to adjust the imaging area so as to include the entirety of the neuropil but not too much additional space.

## Tracing a motor neuron and all its synaptic inputs

We manually traced motor neuron DE-3$^R$, a key motor neuron for all the behaviors included in our functional data set. The combined path length of the entire arborization of DE-3$^R$ was 6,109 μm (*Figure 6a,b*; *Figure 7a*). In addition to tracing the neuron, we marked all of its input synapses and then traced each of its presynaptic partners to their somata. Several visually distinct types of synapses were found, among which most prominently: (1) bright terminals with large dark vesicles (*Figure 6c*) and (2) darker terminals with smaller vesicles that occurred mainly in fiber endings and varicosities (*Figure 6d*). The small vesicles are barely resolved in our data set and appear merely as fields of granules. We used TEM on thin slices of a second ganglion to confirm our interpretation of

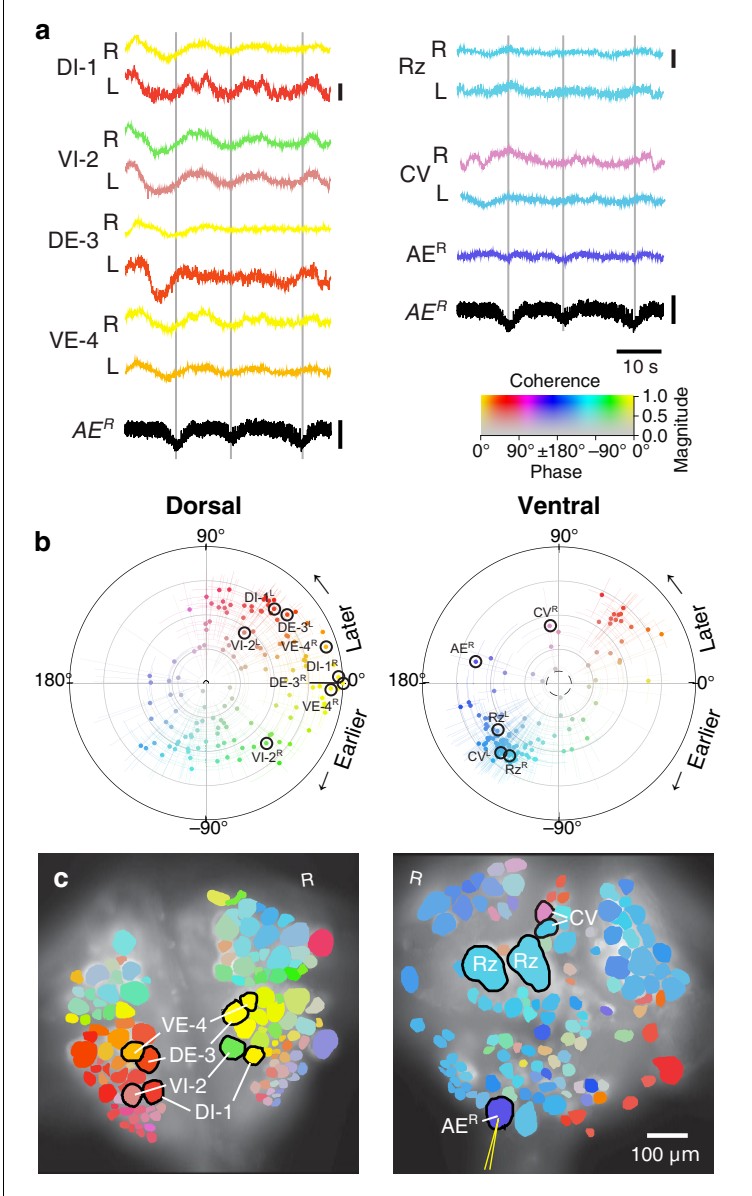

**Figure 3.** Fictive crawling imaged using VSD. (**a**) Selected VSD traces during fictive crawling. From the dorsal surface: dorsal and ventral inhibitory and excitatory motor neurons DI-1, VI-2, DE-3, and VE-4; from the ventral surface: the Retzius cells and CV cells. All those cells are known to be rhythmically active during crawling. Below the VSD traces, a simultaneously recorded intracellular trace of the annulus erector motor neuron AE$^R$ is displayed (in both columns). Scale bars: 0.2%, 10 mV. Gray lines mark hyperpolarized phase of AE$^R$. (**b**) Magnitude and phase of the coherence of activity in individual neurons with the crawl rhythm in motor neuron DE-3$^R$. (**c**) Coherence maps of the VSD signals of all cells on the dorsal (*left*) and ventral (*right*) surfaces of the ganglion. Colors of cell bodies indicate coherence relative to DE-3$^R$. The yellow needle on AE$^R$ indicates a sharp electrode for intracellular recording. Color scale applies to all panels.

these granules as vesicles (*Figure 6—figure supplement 1*). No attempt has been made as of yet to interpret the anatomically distinct types of synapses as physiological classes.

We identified 531 synapses onto DE-3$^R$. Of these, 44 were formed by cells with somata in neighboring ganglia that were not included in our EM volume (*Figure 6—figure supplement 3*). Of the rest, 387 could be traced to their somata with a high degree of confidence. To avoid false positives, we only considered presynaptic neurons that formed at least two synapses onto DE-3$^R$. There were 51 of those. Of those, 35 could be confidently matched to cell bodies seen in the VSD record, and

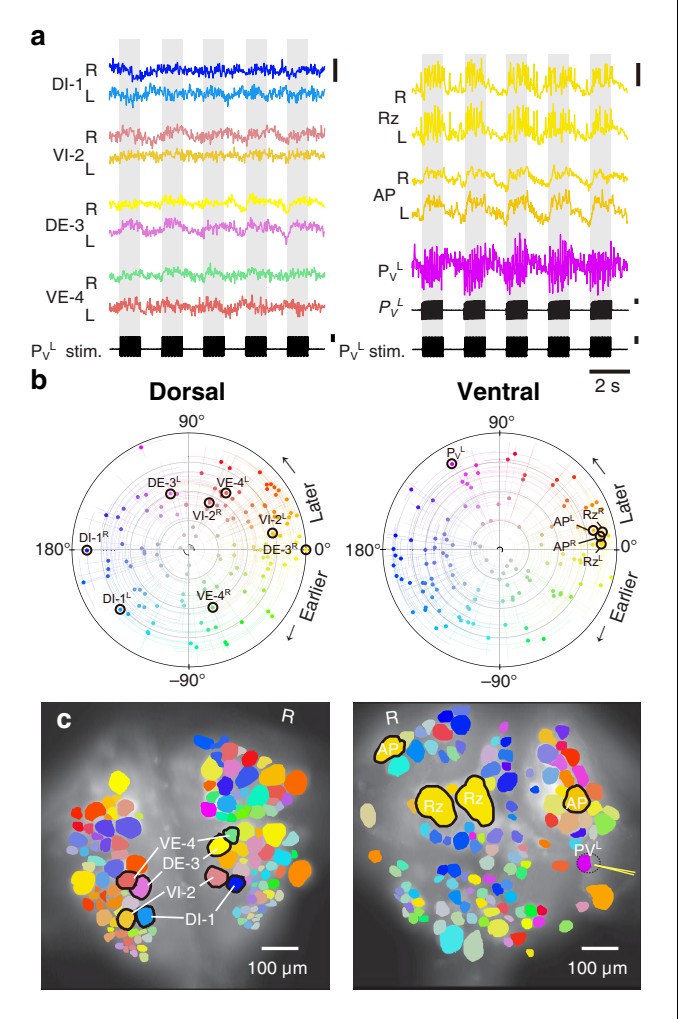

**Figure 4.** Fictive local bending imaged using VSD. (**a**) Selected VSD traces during fictive local bending. From the dorsal surface: dorsal and ventral inhibitory and excitatory motor neurons DI-1, VI-2, DE-3, and VE-4; from the ventral surface: the Retzius cells, 'Anterior Pagoda' cells ('AP'; well-known postsynaptic partners of the P cells with unknown function). Below the traces, a simultaneously recorded intracellular trace of the $P_V^L$ cell is displayed with injected current trains (in both columns). Scale bars: 0.2% relative fluorescence change, 100 mV membrane potential change, 2 nA injected current. (**b**) Magnitude and phase of the coherence of activity in individual neurons with the local bend rhythm in DE-3$^R$. (**c**) Coherence maps of the VSD signals of all cells on the dorsal (*left*) and ventral (*right*) surfaces of the ganglion. Colors of cell bodies indicate coherence relative to DE-3$^R$. A yellow needle on $P_V^L$ indicates a sharp electrode for electrical stimulation. Note that $P_V^L$ was only weakly stained by the VSD because it was left partially covered with sheath tissue to preserve its health.

of those, 10 could be confidently matched to specific identified neurons on the canonical map with previously described functions (*Figure 7b* and *Table 1*). For the others, we assigned previously unused cell numbers from the canonical map (*Wagenaar, 2017*) based on soma location and size (*Figure 7c*). (If there were no unused cell numbers in the vicinity, we reused a previously used cell number but placed a question mark in the figure to indicate that we do not know if our cell and the previously described cell are the same.) The figure also shows some of the cells that we could not confidently link to the VSD record. We did not assign preliminary numbers to those. Our complete tracing results of DE-3$^R$ and its synaptic partners are shown in *Figure 8* and *Video 1*.

## Linking form to function

The availability of both functional (VSD) and anatomical (SBEM) information from the same individual ganglion allowed us to ask questions that relate form to function. First we asked whether there was

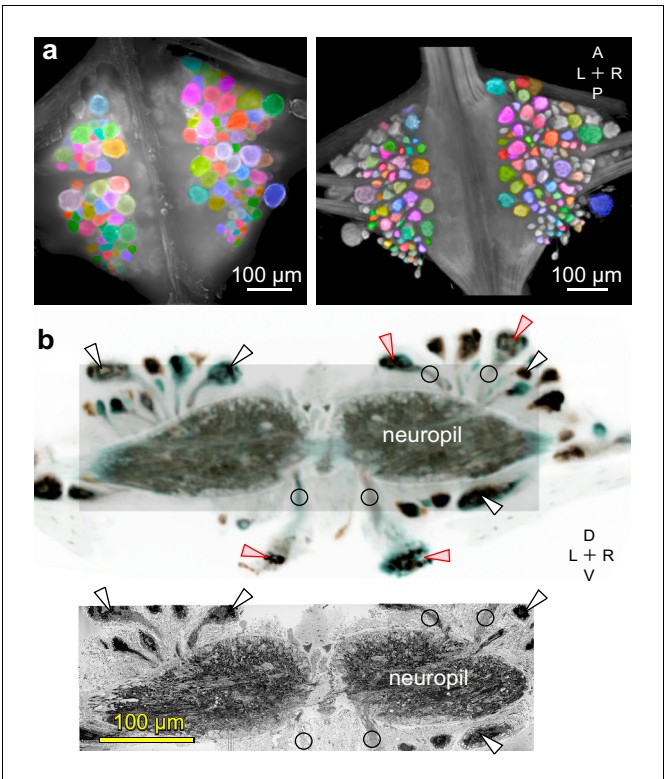

**Figure 5.** Mapping between functional and anatomical images. (**a**) Light micrograph (*left*) and X-ray image (*right*) of the ganglion in which we recorded neuronal activity using a VSD. Matching colors label the same cells on the two images. A: Anterior, P: Posterior, L: Left, R: Right. (**b**) Transverse section from X-ray tomographic image stack of the ganglion (*top*). Only the shaded area was imaged with SBEM (*bottom*). Arrowheads mark several somata that are (*white*) or are not (*red*) included in the SBEM volume. Circles mark neurites that facilitated complete mapping between the two imaging modalities. D: Dorsal, V: Ventral. Note that the X-ray image in (**b**) is shown in reverse video relative to the x-ray image in (**a**) for easier visual comparison with the SBEM image.

a relationship between the overal functional strength of involvement of presynaptic partners and the number of synapses they make onto DE-3[R]. We thus calculated the correlation coefficients between synapse count and the previously obtained coherence magnitudes of the synaptic partners in each of the behavioral trials (see *Figures 2*, *3* and *4*). On average across the eight trials in our data set, this correlation was $0.16 \pm 0.12$ (mean $\pm$ SD; $t = 3.41$; $p = 0.011$; two-tailed t-test), even though none of the individual correlation coefficients was statistically significant (*Figure 9*). To test whether these results were robust, we replaced the synapse count by a 'proximity weight' (see Materials and methods) and repeated the analysis. This yielded similar results: The average across trials of the correlation coefficients was again significantly positive (two-sided t-test, $t = 2.77$, $p = 0.028$, $n = 8$).

Next, we asked whether synapses with different valences (excitatory or inhibitory) were differently distributed along the arbors of DE-3[R]. Excitation and inhibition rely on different physiological processes and have asymmetric effects on cell membrane potential. Accordingly, one might expect that excitatory and inhibitory cells synapse onto their target cells in distinct spatial patterns. For instance, to achieve shunting inhibition, inhibitory synapses would have to be located close to the target cell's spike initiation zone. We therefore considered all the input synapses onto DE-3[R] from neurons with previous descriptions in the literature, and annotated them as either excitatory or inhibitory (*Figure 10a*). Presynaptic neurons for which no previous description as excitatory or inhibitory was found were annotated as 'unknown'. The spatial distributions of excitatory and inhibitory synapses were not found to be different from the other ('unknown') synapses, either when distance was measured to the soma (*Figure 10b*), or when it was measured to the primary neurite (*Figure 10—figure supplement 1a*) of the postsynaptic tree.

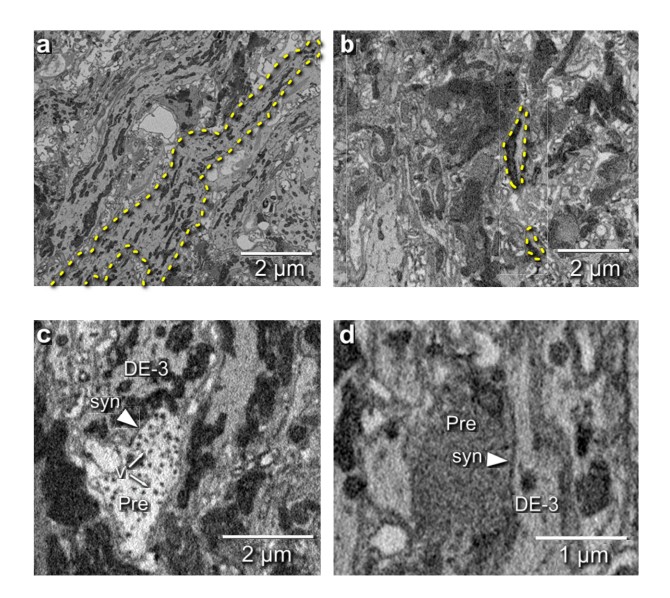

**Figure 6.** Electron microscopic tracing: neurites and synapses of motor neuron DE-3[R]. (**a**) The principal neurite of DE-3[R] near its entrance to the neuropil (*dashed yellow outline*). (**b**) Two branches of the neurite of DE-3[R] (*dashed outlines*). (**c**) A synaptic connection onto DE-3[R] from an inhibitory motor neuron (DI-1[R]). *Arrowheads*: synapses, *Pre*: presynaptic terminal, *v*: vesicles. (**d**) A synapse onto DE-3[R] from an interneuron (cell 24 on the canonical map [*Wagenaar, 2017*]).

The online version of this article includes the following figure supplement(s) for figure 6:

**Figure supplement 1.** Comparison of SEM with TEM for interpreting synapses.

**Figure supplement 2.** Basis for estimating true resolution of SEM images: Spectral power in the images.

**Figure supplement 3.** Workflow for identifying neurons in our data with the canonical map of the ganglion.

In the same vein, we started from the premise that motor neuron DE-3[R], as an output neuron of a multifunctional circuit, must integrate a diverse set of inputs in a flexible manner to accurately generate distinct behaviors. This versatility requires a dynamic functional reorganization of the underlying structural circuit. The question hence arises what the principles governing this dynamic reorganization are. We considered whether the anatomy of DE-3[R] facilitates reading out the different patterns of synchronicity in its presynaptic partners during different behaviors.

We first looked at all neurons that we could cross-reference between EM and VSD recordings (regardless of whether the function of those neurons had previously been described) and, for each of the three behaviors, selected the cells that exhibited the highest coherence with DE-3[R] in that behavior (see Materials and methods). We asked whether cells associated in that way with a specific behavior would form synapses in specific locations, but found that was not the case at the macroscopic scale (*Figure 10c,d* and *Figure 10—figure supplement 1b*).

The absence of an obvious modular organization of DE-3[R] at the cellular scale leaves open the possibility of structure at the synaptic scale exists that relates to the different behaviors. To test that idea, we asked whether synchronously active presynaptic partners form synapses onto DE-3[R] that are spatially clustered. Such spatial clustering of synapses of synchronized cell assemblies has been previously observed in other model organisms (*Takahashi et al., 2012*; *Varga et al., 2011*).

Our spatial clustering algorithm had two free parameters: the maximum allowable distance between nearest neighbor synapses ($d_{NN}$) and the maximum overall cluster extent ($d_{ext}$); see inset to *Figure 11b* and Materials and methods. Findings from in vivo and in vitro studies have shown that neighboring synapses that are less than 12–16 µm apart are more likely to be synchronized than synapses farther apart (*Kleindienst et al., 2011*; *Winnubst et al., 2015*) and that local synaptic plasticity mechanisms act on similar spatial scales (5–10 µm; *Harvey and Svoboda, 2007*). We used these findings to delineate biologically plausible ranges for our parameters: We explored maximum nearest-neighbor distances ($d_{NN}$) between 5 and 25 µm, and maximum spatial extents ($d_{ext}$) between 10

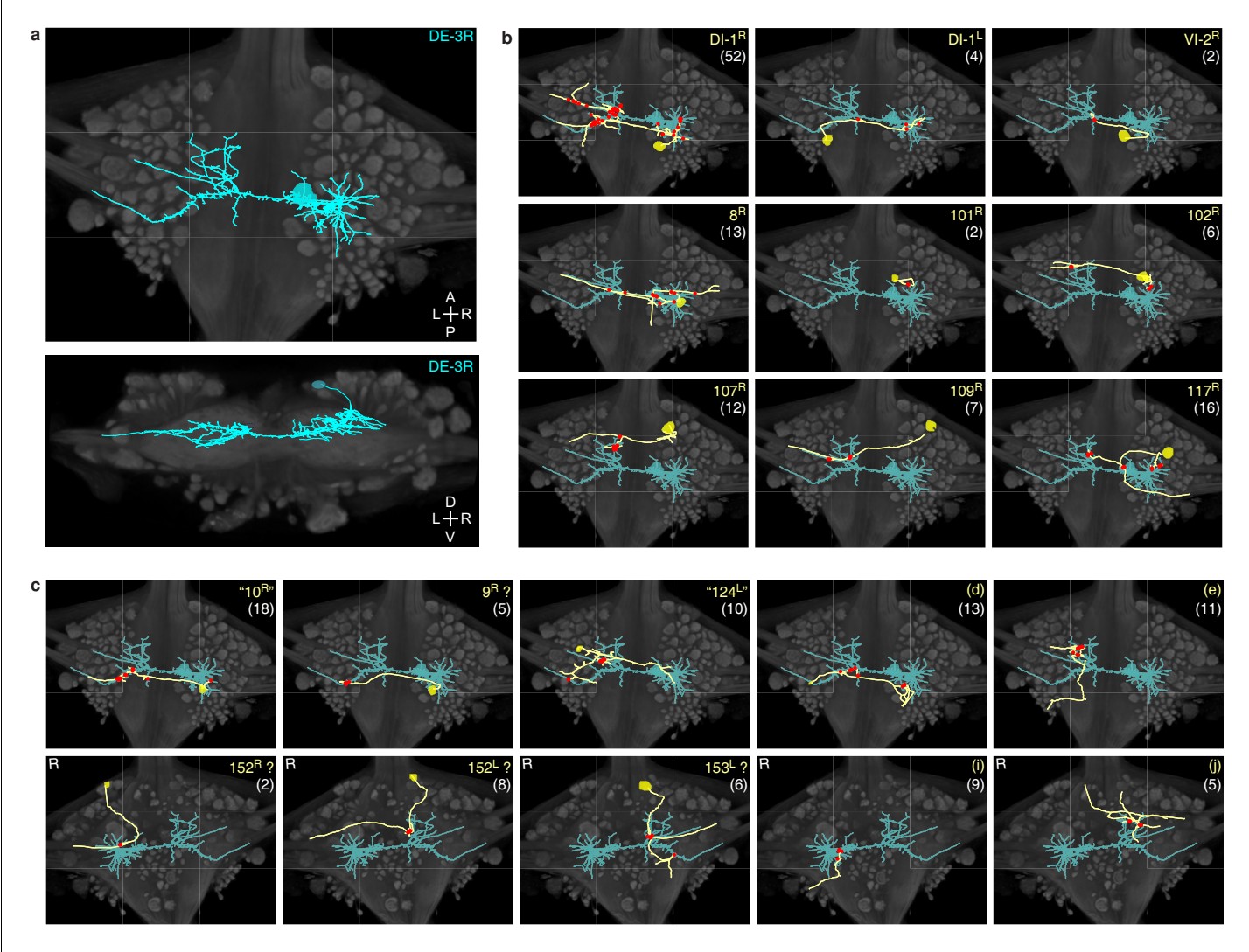

**Figure 7.** Traced neuronal arborizations. (a) Fully reconstructed arborization of DE-3[R] overlaid on a dorsal projection and a transverse section of the micro-CT data. (b) A selection of presynaptic partners with previously known identities. (c) A selection of presynaptic partners not matched to neurons previously described in the literature. (Top row: cell bodies on dorsal aspect; bottom row: cell bodies on ventral aspect of the ganglion.) Other cells in this category are 20[R], 25[R], 154[R], 156[L]. Numbers in parentheses are number of synapses between each cell and DE-3[R].

and 100 µm; the extended upper bounds relative to the literature allowed for a thorough assessment of possible clusters of synapses with synchronized activity.

At all points in this parameter space, the algorithm identified a multitude of synaptic clusters on the neurites of DE-3[R]. Most of these clusters contained synapses from multiple partner neurons (*Tables 2* and *3*).

Since the mere observation of spatial clusters does not demonstrate their functional relevance, we searched through the parameter space of the clustering algorithm to find parameter values that resulted in clusters in which the participating neurons shared commonalities in their activity during various behaviors. This was quantified as an 'F-ratio' (see Materials and methods) that captured the degree to which neurons in a spatial cluster also formed functional clusters in the coherence plot for a given behavior (*Figure 2c*, *Figure 3c*). The overall procedure is outlined in *Box 1*.

In all but one trial, parameter ranges could be identified for which spatial clusters indeed corresponded to functional groupings (*Figure 11b*). We used a least-squares fit approach to find the location in parameter space of the strongest correspondence (*Figure 11c*, *Figure 11—figure supplement 2*, and Materials and methods). In the two swim trials, the peaks were located at $d_{ext}$ =

**Table 1.** Identified partner neurons of DE-3$^R$.
These synaptic partners could be confidently assigned as previously described neurons.

| Cell | Synapse count | Known function |
|---|---|---|
| DI-1$^L$ | 4 | Inhibitor of dorsomedial longitudinal muscles |
| DI-1$^R$ | 52 | Inhibitor of dorsomedial longitudinal muscles |
| VI-2$^R$ | 2 | Inhibitor of ventral longitudinal muscles |
| 8$^R$ | 13 | Excitor of ventral longitudinal muscles |
| 101$^R$ | 2 | Inhibitor of dorsoventral muscles |
| 102$^R$ | 6 | Inhibitor of dorsal longitudinal muscles |
| 107$^R$ | 12 | Excitor of dorsomedial longitudinal muscles |
| 109$^R$ | 7 | Excitor of lateral dorsoventral muscles |
| 117$^R$ | 16 | Excitor of medial dorsoventral muscles |
| L$^R$ | 3 | Excitor of dorsal and ventral longitudinal muscles |

$61 \pm 2$ µm and $65 \pm 2$ µm respectively; in the two crawl trials at $14 \pm 15$ µm and $18 \pm 3$ µm. In summary, functionally relevant spatial clusters during swimming were governed by very different parameter values than during crawling.

## Discussion

We combined voltage-sensitive dye imaging with serial blockface electron microscopy to obtain a comprehensive recording of neuronal activity at the scale of an entire leech ganglion along with a full record of the anatomy of that same ganglion.

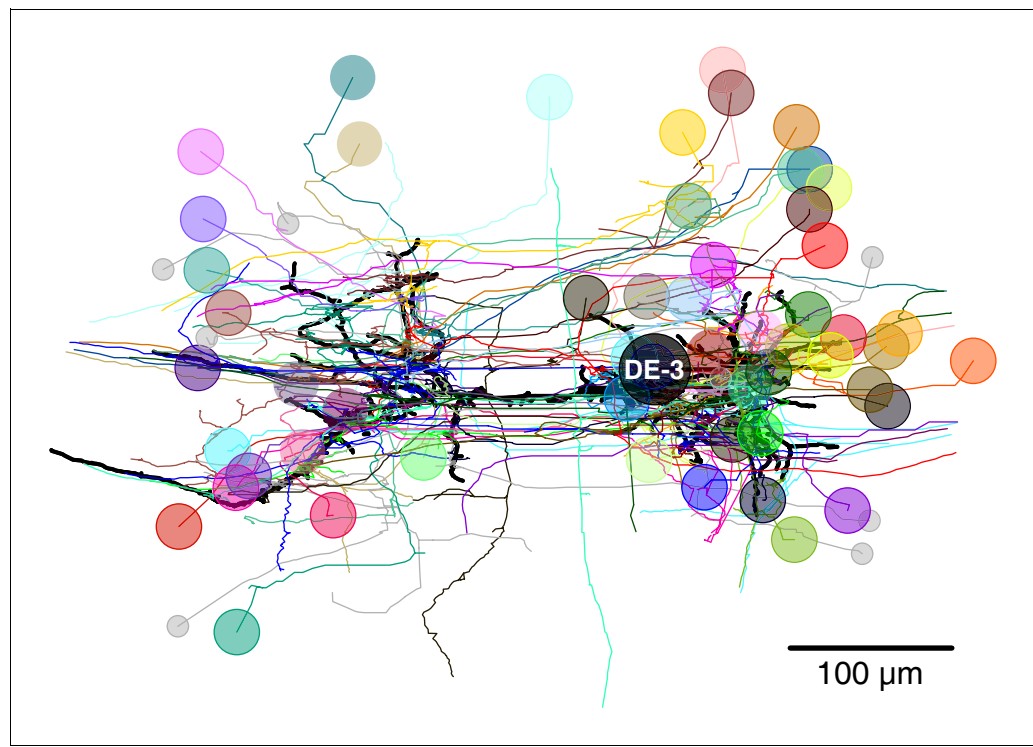

**Figure 8.** Full tracing of DE-3$^R$ (thick black line, soma location marked 'DE-3') and backtracings of all synaptic partners. Partners that we could identify with cells in the VSD recording are marked with (arbitrary) colors. Small gray disks indicate partner neurons that could not be cross-identified between EM and VSD image stacks.

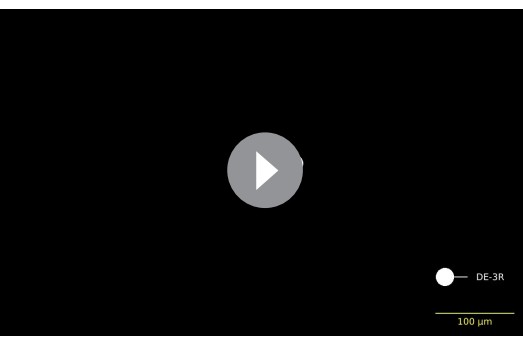

**Video 1.** Full tracing of motor neuron DE-3^R with all of its input synapses, visualized growing out from the soma to the distal branches (0:00 to 0:05). Back tracings of all DE-3^R's presynaptic partners to their somata (0:05 to 0:12). Rotational view of the completed tracing (0:12 to 0:15).

https://elifesciences.org/articles/61881#video1

The use of a fast and sensitive VSD (*Woodford et al., 2015*) allowed us to record even small membrane potential fluctuations from somata. Those signals included both subthreshold excitatory and inhibitory synaptic potentials that could not have been detected by calcium imaging. Conversely, SBEM (*Denk and Horstmann, 2004*) allowed us to image the entire neuropil with sufficient resolution to visualize even thin neurites through much of the volume.

Within this vast dataset, we have focused on a neuron that plays three very distinct dynamic roles in different behaviors: motor neuron DE-3^R, a main excitatory motor neuron of the dorsal longitudinal muscle, as well as its presynaptic partners (*Stent et al., 1978*). We reconstructed its arborization and traced its presynaptic partners back to their somata (up to the limits enabled by our SEM images, see *Figure 6* and supplements). Thus, we generated a detailed map of the output stage of a multifunctional circuit that controls many of the animal's main forms of gross body movement, including swimming, crawling, and the local bend escape.

The reconstructed morphology of DE-3^R was in accordance with previous light microscopic studies in adult (*Norris and Calabrese, 1987*; *Fan et al., 2005*) and electron microscopy in a juvenile ganglion (*Pipkin et al., 2016*): its primary neurite emerged from the soma laterally and traveled toward the ipsilateral roots before making a 180˚ turn to run laterally across the ganglion. Our tracing revealed 531 synapses onto DE-3^R. This was in the same range as the numbers previously reported for a juvenile ganglion (437 and 650) (*Pipkin et al., 2016*). Unlike *Pipkin et al., 2016*, we did not find any output synapses from DE-3^R onto other cells in the ganglion. Several of the input synapses we found were from presynaptic partners that had previously been reported (*Ort et al., 1974*; *Wagenaar, 2017*) based on paired electrophysiological recordings. We focused here on chemical synaptic connections, since SBEM does not yet allow for the identification of gap junctions. However, electrical synapses between DE-3^R and their contralateral homologs as well as some interneurons do exist (*Fan et al., 2005*). Work on molecular markers of electrical synapses that do not require expression of a particular connexin (*Shu et al., 2011*) may overcome this hurdle in the foreseeable future.

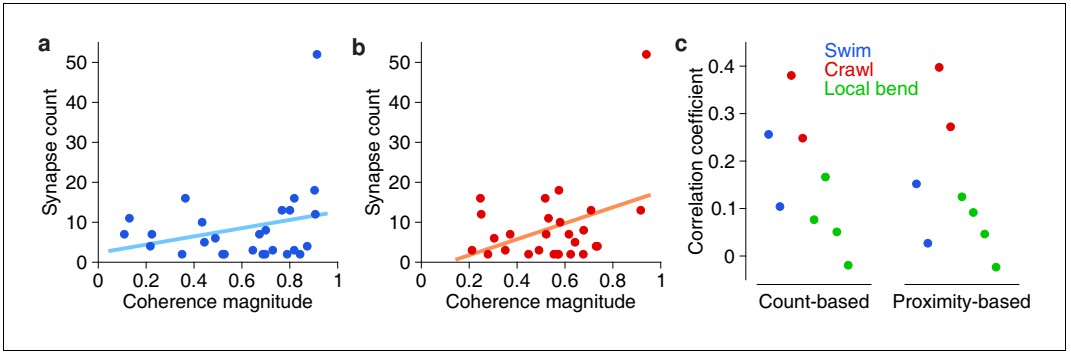

**Figure 9.** Correlation between anatomical and functional measures of synaptic strength. (a) Scatter plot of synapse count vs coherence magnitude during swimming for presynaptic partners with at least two synapses, with linear regression line (data from swim trial #1, $R = 0.26$, $p = 0.20$, n.s.). (b) Same for crawling (data from crawl trial #1, $R = 0.38$, $p = 0.05$, n.s.). (c) Correlation coefficients for all trials (*left*) and when raw synapse count was replaced by proximity weight (*right*; see Materials and methods).

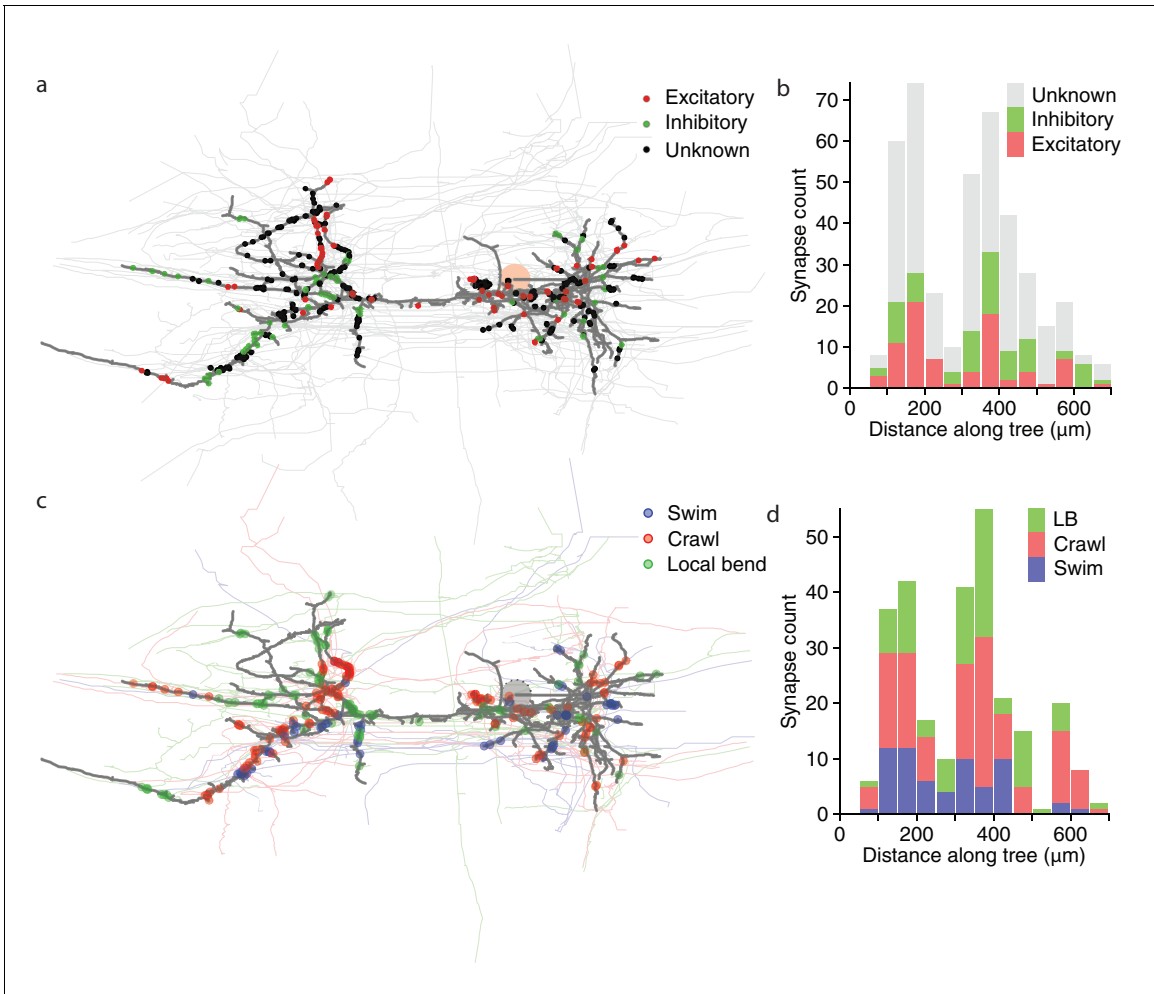

**Figure 10.** Spatial distribution of synapses onto DE-3. (**a**) Distribution of excitatory and inhibitory synapses. (**b**) Histogram of the length of the paths between those synapses and the soma. (**c**) Distribution of synapses more strongly associated with a certain behavior. (**d**) Histogram of the length of the paths between those synapses and the soma (LB: Local bend).

The online version of this article includes the following figure supplement(s) for figure 10:

**Figure supplement 1.** Path lengths between synapses and the trunk of DE-3[R].

In previous reports, 21 neurons (including bilateral homologues) were found to be monosynaptically connected to the DE-3[R] motor neuron (*Figure 12*). Of those, four (bilateral DI-1, 107, and L) were matched to the presynaptic partners seen in the VSD imaging and identified with sufficient confidence. The other neurons previously reported were not confidently matched to presynaptic partners found here. Six other new partners were confidently identified (VI-2, 8, 101, 102, 109, and 117), and a further 41 synaptic partners of DE-3[R] could not be confidently identified with any previously described neuron, mainly because our limited tracing did not allow a morphological identity match. With additional tracing, many of those remaining neurons could possibly be assigned to other previously reported presynaptic neurons shown in *Figure 12*.

The activity patterns observed during all three fictive behaviors exhibited by the single leech in this study (*Figures 2*, *3* and *4*) matched observations from a larger group of animals reported before (*Tomina and Wagenaar, 2017*): The majority of neurons in the ganglion were phasically active during multiple behaviors, indicating substantial but not complete overlap between the circuits that govern the behaviors (*Briggman and Kristan, 2006*). There is considerable variation in the coherence values of specific cells between animals, but the data here fall within the range of values seen before. We conclude that the leech used in this study was representative of its species.

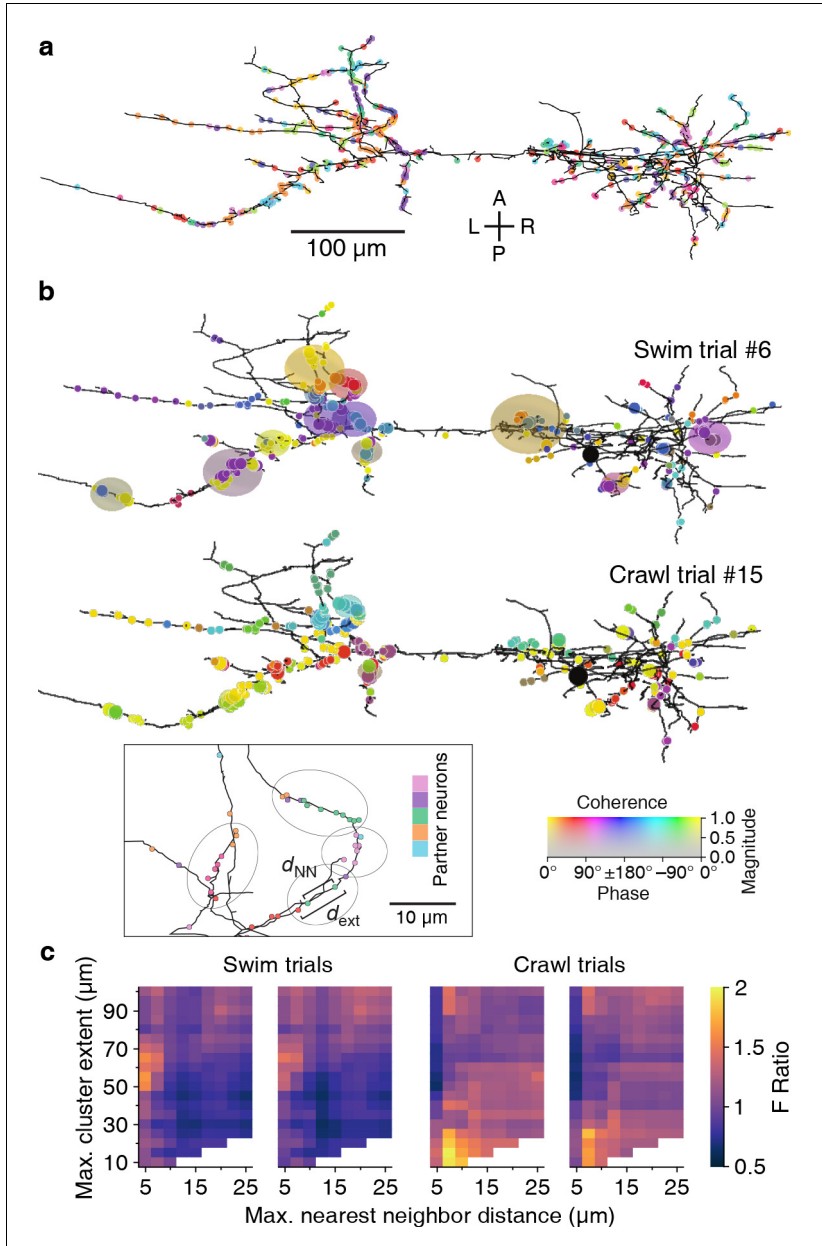

**Figure 11.** Synaptic clustering. (a) Tracing of DE-3[R] with synapses (arbitrarily) colored by presynaptic partner. (b) Clusters (elliptic areas) associated with synchronized synapses for a swimming trial (top) and a crawling trial (bottom). Within the same connectome, clusters of synchronized synapses differ with respect to their spatial extent for the two behaviors: During swimming, synchronization extends over larger areas along the neurite than during crawling. Shown are synaptic clusters obtained with parameter values $(d_{NN}, d_{ext})$ = (5 µm, 65 µm) for swimming and $(d_{NN}, d_{ext})$ = (7.5 µm, 15 µm) for crawling, respectively. Synapses are colored by the coherence between the activity of their presynaptic partner and DE-3[R] during the behaviors (as in *Figure 2*) and clusters are colored by the average coherence of their constituent presynaptic partners. Inset: Explanation of clustering parameters (see Materials and methods). (c) All clustering results for swim and crawl trials. Color indicates the degree of correspondence between spatial clusters and functional grouping expressed as an F-ratio from complex ANOVA (see Materials and methods) as a function of clustering parameters.

The online version of this article includes the following source data and figure supplement(s) for figure 11:

**Source data 1.** Results of the ANOVA analysis of synaptic clusters in all trials.
**Figure supplement 1.** Clustering results for the local bend trials.
**Figure supplement 2.** Peak F-ratios for connecting spatial clusters to functional activity.
**Figure supplement 3.** Demonstration of the F-ratio method using synthetic data.

**Table 2.** Frequency of clusters of different sizes for clustering parameters ($d_{NN}$, $d_{ext}$) = (5 μm, 65 μm), the parameters that gave the largest F-ratio for the swim trials.
*Synapse count:* Number of synapses in a cluster. *Frequency:* Number of clusters with the respective synapse count. *Number of presynaptic partners:* Number of unique presynaptic neurons contributing to the synapses in the respective clusters.

| Synapse count | Frequency | Number of presynaptic partners |
|---|---|---|
| 2 | 9 | 1 (6x), 2 (3x) |
| 4 | 4 | 1 (1x), 2 (3x) |
| 5 | 1 | 1 (1x) |
| 6 | 2 | 2 (2x) |
| 7 | 1 | 2 (1x) |
| 8 | 2 | 2 (1x), 3 (1x) |
| 9 | 3 | 3 (1x), 4 (2x) |
| 11 | 1 | 3 (1x) |
| 15 | 1 | 3 (1x) |
| 23 | 1 | 6 (1x) |

Overlaying the anatomical and activity images (*Figure 5*) and comparing with the established canonical maps of the ganglion (*Wagenaar, 2017*) allowed us to address form–function relationships. For instance, our own data allowed us to ask whether neurons that are more strongly associated with a particular behavior than with others formed synapses onto the output motor neuron that were spatially localized. Such an arrangement would indicate a modular organization of the motor neuron's processes (*London and Häusser, 2005*). However, we found no evidence of such organization in DE-3[R] (*Figure 10c,d*) other than a slight trend that swim-associated synapses tended to be located closer to the primary neurite (*Figure 10—figure supplement 1b*). Since most presynaptic partners are multifunctional, the pre-motor network undoubtedly plays an important role in generating distinct motor patterns.

We also asked whether synapses with a particular valence (excitatory or inhibitory) were differently distributed along the tree, which would likewise have implications for possible models of computation in the cell (*Saha and Truccolo, 2019*). Thanks to an extensive body of previously published electrophysiological recordings from the leech ganglion, the valence of many of DE-3[R]'s presynaptic partners is known, allowing us to visualize the spatial distributions of excitatory and inhibitory synapses separately (*Figure 10a*). However, we found no obvious differences between the two classes of synapses (*Figure 10b*, *Figure 10—figure supplement 1a*).

The lack of macroscopic organization of functionally related synapses does not imply that the spatial arrangement of synapses on the tree has no functional relevance. Indeed, the limited

**Table 3.** Frequency of clusters of different sizes for clustering parameters ($d_{NN}$, $d_{ext}$) = (7.5 μm, 15 μm), the parameters that gave the largest F-ratio for the crawl trials.
Columns as in *Table 2*.

| Synapse count | Frequency | Number of presynaptic partners |
|---|---|---|
| 2 | 28 | 1 (23x), 2 (5x) |
| 3 | 11 | 1 (7x), 2 (4x) |
| 4 | 9 | 1 (4x), 2 (5x) |
| 5 | 2 | 1 (2x) |
| 6 | 2 | 2 (2x) |
| 7 | 3 | 1 (1x), 2 (1x), 3 (1x) |
| 12 | 1 | 3 (1x) |

> ## Box 1. Clustering procedure.
>
> 1. Select pairs of values from the parameter space $5\ \mu m \leq d_{NN} \leq 25\ \mu m$ and $10\ \mu m \leq d_{ext} \leq 100\ \mu m$.
> 2. For each such pair $(d_{NN}, d_{ext})$:
>     a. Find all synaptic clusters on DE-3$^R$ based on path-based distances;
>     b. Remove clusters with synapses from only one presynaptic partner or a single synapse.
> 3. For the given clusters and for each behavioral trial:
>     a. Calculate the sum of squares of the coherence values within and between clusters;
>     b. Calculate the F-ratio based on the sum of squares.

experimental evidence currently available suggests that even clusters of two synapses are functionally relevant in the mammalian cortex (*Takahashi et al., 2012*; *Fu et al., 2012*). Findings from computational (*Poirazi and Mel, 2001*) and in vitro studies (*Losonczy and Magee, 2006*; *Nevian et al., 2007*) suggest that individual dendritic branches act as integrative compartments and that spatial synaptic clusters facilitate nonlinear dendritic integration. These results imply that connectivity has to be specific with respect to individual dendritic branches. Such a precision could arise through spontaneous neural activity where synaptic clusters are established by a branch-specific 'fire-together-wire-together' rule (*Kleindienst et al., 2011*).

We therefore applied a spatial clustering algorithm to the input synapses on DE-3$^R$ and asked whether the clusters identified by that algorithm also stand out as functionally significant groupings. Indeed, we found a set of parameter values for which synapses within spatial clusters also clustered in the phase space of the swim rhythm and a different set of parameters for which synapses within spatial clusters also clustered in the phase space of the crawl rhythm (*Figure 11b,c*). Results were highly consistent between trials of the same behavior. (The results of clustering were not consistent among local bending trials. This may stem from a nature of this reflex behavior: the responsiveness of the population of neurons involving pressure sensation is variable, reflecting the plasticity of the circuit (*Crisp and Burrell, 2009*).)

Earlier work relied on light microscopy and could therefore not identify presynaptic partners. In contrast, our use of SBEM allowed us to assess such connection specificity based on individually identified synapses. Our findings confirm previous observations on the synchrony of proximal synapses (*Kleindienst et al., 2011*; *Takahashi et al., 2012*). An attractive interpretation of our results is that the clusters are the loci where inputs from synchronized presynaptic cell assemblies are integrated (*Briggman and Kristan, 2008*). In agreement with earlier reports (*Kleindienst et al., 2011*), the strongest correspondence between spatial clusters and activity patterns was observed when synaptic clusters were defined by a maximum distance between nearest neighbors ($d_{NN}$) of up to 10 μm.

Computational and in vitro studies indicate that 10–20 synchronized inputs are required to trigger dendritic spikes (*Ujfalussy and Makara, 2020*). Existing in vivo studies, however, have found synchrony in only up to five neighboring dendritic spines, making it unclear what the impact of the synaptic clusters on the membrane potential of the posysynaptic neuron might be. The smaller number derived from in vivo conditions might stem from the limitations of experimental methods. Indeed, our results indicate that clusters of up to 23 synaptic members can display synchronous activity. Further work is required, however, to elucidate the role of synaptic clusters in vivo or ex vivo on the membrane potential of the postsynaptic neuron under different experimental conditions.

Additional studies complementing our data are also required to more fully understand the mechanisms underlying synaptic integration in motor neurons like DE-3$^R$. For instance, single-cell voltage imaging (*Kuhn and Roome, 2019*) and voltage clamp using sharp electrodes on neurites (*Laurent, 1993*; *Takashima et al., 2006*) would help determine where in the neurites active propagation of membrane potentials is supported and thus lead to a better understanding of how postsynaptic

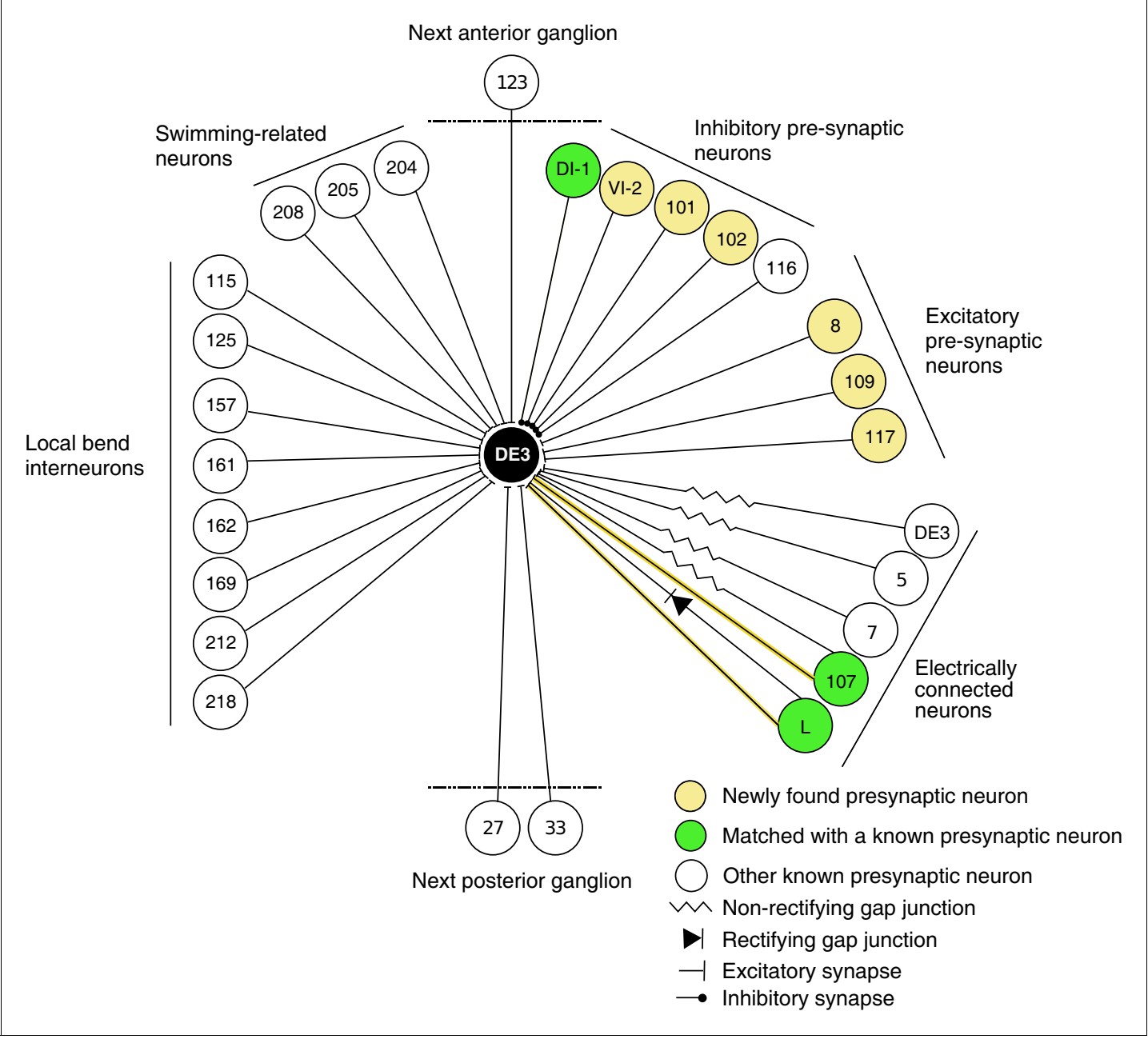

**Figure 12.** Wiring diagram of DE3. The diagram shows synaptic connections onto DE3, including newly found presynaptic neurons in this study (yellow) and previously known ones (green if also found in this study; white otherwise). Only confidently identified neurons shown in *Figure 7* and *Table 1* are represented as 'found.' Other presynaptic partners found but not positively identified in this study are not shown here. This diagram does not depict whether connections are ipsi- or contralateral.

potentials as well as action potentials propagate through the tree. As it stands, we only have voltage data from the soma. Leech neurons—along with most neurons in the central nervous systems of other annelids, arthopods, and molluscs—have a unipolar morphology with the somata strictly segregated from neuropil (*Bullock and Horridge, 1965*). With notable exceptions (including sensory neurons in the leech and select insect motorneurons [*Hancox and Pitman, 1992*]), these somata are thought to play a relatively minor role in the integration of synaptic inputs to action potential outputs (*Andjelic and Torre, 2005*).

Even so, the completeness of our functional dataset in combination with our anatomical data makes for an attractive basis for simulation studies to arrive at a computational understanding of

multifunctional neuronal circuits (*Real et al., 2017*). Our data may also serve as a large-scale ground truth for EM segmentation algorithms (*Plaza and Funke, 2018*). This study not only represents an important step in a combined approach to studying multifunctional circuits at the synaptic level, but also lays the groundwork for a comprehensive neuronal mapping of a whole ganglion that semi-autonomously processes local sensory information and controls segmental movement.

The combination of anatomical methods with synaptic resolution and imaging techniques that can record from the entirety of the neurons of a circuit promises an extraordinary opportunity to assess neural computations at the level of circuit dynamics. Functional maps (*Alivisatos et al., 2012*) from recorded activity combined with anatomical connectomes (*Denk et al., 2012*; *Bargmann and Marder, 2013*; *Morgan and Lichtman, 2013*) are therefore poised to become a powerful tool not only to have a better understanding of complex behaviors, but also to predict the outcomes of new manipulations *Devor et al., 2013*; *Kandel et al., 2013*; *Wang et al., 2013*. This new addition with the annelid species to the developing field would also pave the way for comparative approaches by functional connectomics to study evolutionary aspects of neuronal design (*Laurent, 2020*).

# Materials and methods

### Key resources table

| Reagent type (species) or resource | Designation | Source or reference | Identifiers | Additional information |
|---|---|---|---|---|
| Strain, strain background (*Hirudo verbana*) | Wild-type background | Niagara Leeches | | |
| Other (voltage-sensitive dye) | VF2.1(OMe).H | *Woodford et al., 2015* | | Courtesy Miller lab |
| Software, algorithm | SBEMAlign | This paper | | https://github.com/wagenadl/sbemalign; *Wagenaar, 2021a*; copy archived at swh:1:rev:d76dcc55e7dad3e7bca91de24d20d201696a5339 |
| Software, algorithm | SBEMViewer | This paper | | https://github.com/wagenadl/sbemviewer; *Wagenaar, 2021b*; copy archived at swh:1:rev:8f8d3d2bcae39e165993d9e11ffe173640b940db |
| Software, algorithm | GVox | This paper | | https://github.com/wagenadl/gvox; *Wagenaar, 2021c*; copy archived at swh:1:rev:5e7ccd2273caed49bac3e09ca39de68a0b182fc5 |

## Dissection and voltage-sensitive dye imaging

Detailed procedures have been described before (*Tomina and Wagenaar, 2018*). Briefly, leeches (*Hirudo verbana*, obtained from Niagara Leeches, Niagara Falls, NY) were maintained on a 12 hr:12 hr light:dark cycle in temperature-controlled aquariums filled with artificial pond water. The entire nervous system of an adult leech was removed and pinned down on silicone (PDMS, Sylgard 184, Dow Corning, Midland, MI). The sheath surrounding one segmental ganglion (M10) was removed from both ventral and dorsal aspects to allow access with voltage-sensitive dyes. Most of the nerves that innervate the periphery were cut short, but several were kept long to allow extracellular stimulation as described before (*Tomina and Wagenaar, 2017*). A voltage-sensitive dye (VF2.1(OMe).H [*Woodford et al., 2015*] provided by Evan Miller) was bath-loaded at a concentration of 800 nM in leech saline using a pair of peristaltic pumps to evenly load cell membranes on both sides of the ganglion. The preparation was placed on a custom-built dual-headed microscope which was used to image neuronal activity during fictive behaviors triggered by electrical stimulation, as in our previous work (*Tomina and Wagenaar, 2017*).

We manually drew regions of interest (ROIs) around neuronal cell bodies and used custom software to associate those ROIs with named cells on the canonical maps of the leech ganglion (*Wagenaar, 2017*). For each of the behavior trials separately, we calculated the spectral coherence between each of the neurons and DE-3$^R$ at the frequency of the dominant peak in the power spectrum of DE-3$^R$ for the given behavior.

## Histology

After dye imaging, the preparation was reduced to just one segmental ganglion by transecting the anterior and posterior connectives. The ganglion was mounted on a slab of silicone (DPMS) with a

hole cut out in the center so that the somata would not be in direct contact with the silicone. This preparation was transferred into a glass container and incubated for 72 hr at 4 °C in 2% paraformaldehyde, 2.5% glutaraldehyde in 0.15 M cacodylate buffer containing 2 mM $CaCl_2$. Subsequently, the ganglion was washed in cacodylate buffer for 10 min and then incubated in an aqueous solution of 2% $OsO_4$ and 1.5% potassium ferrocyanide. During this incubation, the sample was microwaved in a scientific microwave (Pelco 3440 MAX) three times at 800 W with a duty cycle of 40 s on and 40 s off at a controlled temperature of 35 °C and subsequently left at room temperature (RT) for 30 min. The sample was then washed twice in $ddH_2O$ and then microwaved three times at 30 °C with a duty cycle of 2 min on and 2 min off.

The sample was incubated in 0.5% thiocarbohydrazide (Electron Microscopy Sciences, Hatfield, PA). During this incubation, the sample was microwaved three times at 800 W with a duty cycle of 40 s on and 40 s off at 30 °C and subsequently left at RT for 15 min. The ganglion was then washed again, followed by the same microwave incubation as described above.

Next, the sample was incubated in 2% aqueous $OsO_4$, microwaved three times at 800 W with a duty cycle of 40 s on and 40 s off at 30 °C, and left for 30 min at RT. After another wash, the sample was left overnight in 2% uranyl acetate at 4 °C.

The next day, the sample was incubated in a lead aspartate solution at 60 °C for 25 min (*Walton, 1979*). The sample was then washed and dehydrated through a series of ethanol solutions (50%, 70%, 90%, 100%, 100%, 10 min each) at RT and incubated in acetone. After this, the sample was infiltrated with epoxy resin by first incubating it for one day at RT in a solution of 25% Durcupan (Sigma, St. Louis, MO) in acetone. On subsequent days, the concentration of Durcupan was increased to 50%, 75%, and finally 100%. After that, the sample was transferred to freshly prepared 100% Durcupan and incubated at 60°C for 3 days.

## Micro-CT imaging

We used Micro-CT scanning to confirm that the above sample preparation had left the overall geometry of the ganglion intact and to trace portions of neurons outside of the neuropil. Scans were collected using the 20x objective on a Zeiss Versa 510 X-ray microscope. Epoxy-embedded ganglia were attached to the end of an aluminum rod using cyanoacrylate glue and then scanned at 80 kV, collecting 2401 projection images while rotating the specimen 360°. The final pixel size was approximately 0.75 µm. Volumes were reconstructed using Zeiss Reconstructor software and visualized in custom software (GVox, see Key resources table).

## Scanning electron microscopy

Ganglia were mounted onto aluminum pins using conductive silver paint. They were mounted in a vertical orientation (with the anterior connective pointing upwards). The sample was imaged with a Zeiss Gemini 300 SEM with a Gatan 2XP 3View system. The microscope was run in focal charge compensation mode (*Deerinck et al., 2018*) using nitrogen gas (40% pressure), with an accelerating voltage of 2.5 kV, a 30 µm objective aperture, magnification of ×336, a raster size of 17,100 × 17,100 pixels, a 5.5 nm pixel size, a 0.5 µm dwell time, and 50 nm section thickness. Stage montaging with an overlap of 8% between tiles was used to cover the complete extent of the ganglion in any given image. The backscatter detector was a Hamamatsu diode with a 2 mm aperture.

Outside of the neuropil, neuronal processes could be traced in the micro-CT scan, which allowed us to reduce the total volume needed to be imaged with SBEM by almost a factor two (*Figure 5b*). Still, at the widest points of the neuropil as many as 7 × 2 tiles (119,700 × 34,200 pixels) were needed at a given z-position.

After approximately every 500 sections, the run was stopped to clear sectioning debris from the diamond knife and prevent contamination of the block-face, diode, or column. The run was also stopped when we reached significantly wider or narrower regions of the neuropil as indicated above. Ultimately, the run was subdivided into 61 subruns. There was only one instance in the run where a significant loss of tissue occurred (approximately 150 nm) following the re-approach of the knife to the tissue after an interruption for clearing debris. Overall, electron microscopy took 7 months of near-continuous imaging.

To quantify true image resolution (as opposed to pixel size), we calculated power spectra of pixel intensities in several 2048 × 2048 pixel regions throughout the volume (*Figure 6—figure*

*supplement 2*). The spectral power in our images exceeded the noise floor set by shot noise at spatial frequencies up to about 20 lines/µm (*Figure 6—figure supplement 2*), corresponding to an effective pixel size of about 25 nm.

## Transmission electron microscopy

Image quality and specimen preservation were verified using an additional ganglion prepared as above, but imaged in ultrathin sections on a conventional transmission electron microscope (JEOL JEM-1200EX, 120 kV, ×12,000–×20,000 magnification).

## Image processing

Images were aligned using custom software ('SBEMAlign,' see Key resources table). First, we reduced the linear resolution of the original images by a factor five. Then we split each image into 5 × 5 sub-tiles and calculated the optimal alignment between each sub-tile and the corresponding sub-tile from the image above using a modified version of SWIFT-IR (*Wetzel et al., 2016*). Likewise, we split the regions of overlap that existing between images of the same slice into five sub-tiles and calculated the optimal alignment between the edges of adjacent images. We used these latter numbers to coarsely align images within each slice and render the first and last slices of each subrun at 1:25 scale, which allowed us to establish regions of correspondence between subruns. Using these procedures, we ended up with 3,430,650 matching pairs of image locations. Because SWIFT-IR matches up entire areas rather than single point pairs, those locations are defined at a much higher resolution than that of the images. Accordingly, alignment information obtained at a scale of 1:5 could be used to align the source images at scale 1:1 without material loss of precision.

Next, we split the full EM volume up into subvolumes of 200 slices with 50% overlap between subsequent subvolumes (Stefan Saalfeld, personal communication) and optimized alignment in each subvolume independently. This was done in three steps: (1) Coarse alignment of all the z-stacks from all of the subruns involved in the subvolume relative to each other; (2) Refinement of this alignment by determining optimal rigid translation of each tile relative to its substack; (3) Further refinement through elastic deformation. This procedure resulted in absolute coordinates for a grid of points in each source image.

We then rendered each slice by linearly combining the placement according to the two subvolumes that incorporated the slice. We divided each slice up into nonoverlapping rectangles and rendered pixels from one source image into each rectangle using a perspective transformation derived from the grid coordinates calculated in the previous step.

The full-resolution stitched volume was then split into tiles of 512 × 512 × 1 voxels and reduced-resolution tiles at 1:2, 1:4, up to 1:256 resolution were produced for faster online visualization.

## Visualization

We developed a custom tool for visualizing the aligned images and for neurite tracing. SBEMViewer (see Key resources table) was used to visualize the slices as they came off the microscope to monitor image quality, and also for purposes of tracing neurites.

## Neurite tracing

We produced a full skeleton tracing of the right DE-3 motor neuron and all of its presynaptic partners from the synapses to their somata. The following criteria were used to identify synapses:

1. Vesicles have to be evident on the presynaptic site;
2. Those vesicles have to be in immediate proximity of the putative synapse;
3. The pre- and postsynaptic cells must have membrane apposition across at least three sections (150 nm).

Because of limited resolution in our SBEM images, synaptic vesicles appear merely as gray granules (*Figure 6c,d*), but fields of such granules were clearly distinct from other gray areas in the SBEM images. Comparison with digitally blurred TEM images (*Figure 6—figure supplement 1*) confirmed this interpretation. We found that granular areas were concentrated in fiber endings and varicosities.

## Correlation analysis of synapse number and coherence

We calculated correlation coefficients for the number of synapses and the strength of functional connectivity quantified by the magnitude of the coherence in each of the behaviors (*Figure 9*). We then repeated this analysis with synapse count replaced by the 'proximity weight' of the synaptic partners. The proximity weight of a synapse was defined as the inverse of its distance to the soma of DE-3$^R$ (measured along the neurite), and the proximity weight of a neuron was defined as the sum of the proximity weights of its synapses onto DE-3$^R$. Neurons were included in the calculation if (1) their somata could be matched between VSD and SBEM images and (2) they had at least two synapses onto DE-3$^R$ (see also *Figure 6—figure supplement 3*). To test for overall significance, a two-tailed t-test was applied to compare the collected correlation coefficients (from $n$ = 8 trials) to the null hypothesis of zero average correlation.

## Tree analysis

To assign neurons to specific behaviors (*Figure 10c,d*), we took all the neurons for which we had a match between anatomy (EM) and activity (VSD). We then used the following procedure which compensates for the fact that the distribution of absolute coherence values is different per behavior for technical reasons resulting from the differences in cycle periods between the behaviors. We first looked at the coherence values in all cells and all behaviors, and assigned the neuron that had the greatest coherence to the behavior in which it had that coherence value. In the second step, we looked at the coherence values in all the other cells and both other behaviors, and assigned the top-cohering cell to a behavior. In the third step, we looked at the coherence values in all remaining cells in the last remaining behavior, and assigned the top-cohering cell to that behavior. In the fourth step, we once again considered all behaviors. In this manner, we continued until all cells had been assigned to a behavior. It should be noted that most cells were active in all behaviors to some degree, and that not all differences in absolute coherence values between behaviors were large. Accordingly, some variability in assigned should be expected across animals.

The primary neurite was defined as the path between the soma and the point where the axon leaves the ganglion through the contralateral dorsal posterior nerve.

## Synaptic clustering analysis

The analysis was based on data from the 45 synaptic partners of DE-3$^R$ for which both anatomical as well as VSD recordings were available. The overall procedure is outlined in *Box 1*.

We defined synaptic clusters using an agglomerative hierarchical clustering algorithm with two parameters:

1. The maximum allowed distance between nearest neighbors ($d_{NN}$);
2. The maximum overall spatial extent of the cluster ($d_{ext}$).

The algorithm began by treating each synapse as an individual cluster. Then, it iteratively joined the two clusters with minimum distance between their most proximal elements ('single-linkage' clustering). However, if a joint cluster would exceed the limit on overall spatial extent ($d_{ext}$), its putative constituents were not joined. Aggregation stopped when no pairs of clusters were left with acceptable nearest-neighbor distance (i.e. less than $d_{NN}$) and acceptable joint spatial extent (i.e. less than $d_{ext}$). All distances were measured along the neurites of DE-3$^R$ rather than by Euclidean metric in the volume. Clusters comprising only a single synapse were not considered for further analysis.

The analysis of functional significance of spatial clusters used an ANOVA-like procedure on the complex spectral coherence values of neurons within clusters relative to DE-3$^R$. As in ANOVA, we calculated sums of squares within and between clusters. Since coherence values are complex numbers, we used the *absolute* square value. The ratio of these sums of squares (the 'F-ratio') is larger than one if coherence values within a spatial cluster tend to be more similar to each other than coherence values between different clusters.

Specifically, if $z_{k,i}$ represent the (complex) coherence values of cell $i$ in cluster $k$, then the centroid of each cluster is $z_k^0 = \frac{1}{n_k} \sum_i z_{k,i}$ and the overall average of coherence values is $z^0 = \frac{1}{N} \sum_{k,i} z_{k,i}$, where $n_k$ is the number of cells in cluster $k$ and $N$ is the total number of cells. The F-ratio is then

$$F = \frac{\sum_{k,i} |z_{k,i} - z^0|^2}{\sum_{k,i} |z_{k,i} - z_k^0|^2}.$$

A demonstration of the method using synthetic data is presented in *Figure 11—figure supplement 3*. We generated three clusters of two-dimensional Gaussian-distributed data, either with centroids at the same location (*Figure 11—figure supplement 3a*), or with the centroids displaced from each other by one standard deviation in different directions (*Figure 11—figure supplement 3b*). One can imagine each of the three colored clouds of dots as corresponding to the coherences of the neurons in three spatially defined clusters, represented on the complex plane. In *Figure 11—figure supplement 3a*, the distribution of coherence values is the same for each cluster, hence $F = 1$; in *Figure 11—figure supplement 3b*, each spatially defined cluster is clearly distinguishable by the coherence values of its neurons, hence $F > 1$.

In standard ANOVA, the F-ratio follows an F-distribution under the null hypothesis. In the complex-valued case, that is no longer true, so we calculated empirical distributions of the F-ratios by randomly shuffling the list of per-neuron coherence values 1000 times. The empirical p-value $\hat{p}$ was then defined as $\hat{p} = \frac{m+1}{N+1}$, where $N = 1000$ is the number of randomizations and $m$ is the number of times the F-ratio from shuffled data exceeded the experimentally observed F-ratio. These p-values are reported in the Data supplement to *Figure 11*.

We generated plots of the F-ratio as a function of the cluster parameters $d_{NN}$ and $d_{ext}$. For each trial, we first determined the value of $d_{NN}$ for which the largest F-ratio was obtained. Then, we fitted a Gaussian of the form

$$F = 1 + A \exp\left(-\frac{1}{2}[d_{ext} - \mu]^2/\sigma^2\right)$$

to the F-ratio as a function of $d_{ext}$ (*Figure 11—figure supplement 2*). The μ-values from those fits and their uncertainties according to least-squares fitting are reported in the text.

## Data availability

The easiest way to access the raw electrophysiology and voltage-dye data as well as the SBEM image data and tracing results used in this paper is through a series of Python modules that we made available at https://github.com/wagenadl/leechem; *Kassraian and Wagenaar, 2021*; copy archived at swh:1:rev:73eee24e387e11c259a3f3fe0bd4e469048b25e6. Included in the package is a file called 'demo.py' that demonstrates the use of the modules as well as several Jupyter notebooks that demonstrate other approaches to data analysis.

*Table 4* lists the VSD trials available for download using the Python modules. The SBEM volume may also be accessed through the Neuroglancer (*Google, 2016*) instance at https://leechem.caltech.edu or by pointing SBEMViewer to https://leechem.caltech.edu/emdata. This server also allows for direct downloading of SBEM image data. The API is documented at https://leechem.caltech.edu/emdata/help.

**Table 4.** List of raw data trials and how they are referred to in the paper.

| Figure | Behavior | Trial no. in paper | Trial no. in raw data |
|---|---|---|---|
| *Figure 2* | Swim | 1 | 6 |
| – | Swim | 2 | 8 |
| *Figure 3* | Crawl | 1 | 15 |
| – | Crawl | 2 | 17 |
| *Figure 4* | Local bend | 1 | 9 |
| – | Local bend | 2 | 10 |
| – | Local bend | 3 | 11 |
| – | Local bend | 4 | 12 |

## Acknowledgements

We thank Evan Miller (Berkeley) for sharing of the VF2.1(OMe).H dye, Art Wetzel (Pittsburgh Super-computer Center) and Tom Bartol (The Salk Institute) for many useful discussions about image alignment, Tünde Magyar and Renáta Pop (University of Veterinary Medicine, Department of Anatomy and Histology, Budapest, Hungary) for help with TEM, and Jason Pipkin (Brandeis University) for advice about tracing.

## Additional information

### Funding

| Funder | Grant reference number | Author |
|--------|------------------------|--------|
| National Institute of Neurological Disorders and Stroke | R01-NS094403 | William B Kristan Jnr<br>Mark H Ellisman<br>Daniel A Wagenaar |
| National Institute of General Medical Sciences | P41-GM103412 | Mark H Ellisman |
| Japan Society for the Promotion of Science | 201800526 | Yusuke Tomina |
| Japan Society for the Promotion of Science | 19K16191 | Yusuke Tomina |
| Swiss National Science Foundation | P2EZP3-181896 | Pegah Kassraian |
| National Institute of Neurological Disorders and Stroke | 1U24NS120055 | Mark H Ellisman |

The funders had no role in study design, data collection and interpretation, or the decision to submit the work for publication.

### Author contributions

Mária Ashaber, Formal analysis, Investigation, Visualization, Methodology, Writing - original draft, Writing - review and editing; Yusuke Tomina, Conceptualization, Data curation, Formal analysis, Funding acquisition, Investigation, Visualization, Methodology, Writing - original draft, Writing - review and editing; Pegah Kassraian, Conceptualization, Software, Formal analysis, Funding acquisition, Investigation, Visualization, Methodology, Writing - original draft, Writing - review and editing; Eric A Bushong, Investigation, Methodology; William B Kristan, Conceptualization, Funding acquisition, Methodology; Mark H Ellisman, Conceptualization, Supervision, Funding acquisition, Methodology, Project administration; Daniel A Wagenaar, Conceptualization, Data curation, Software, Formal analysis, Supervision, Funding acquisition, Visualization, Methodology, Writing - original draft, Project administration, Writing - review and editing

### Author ORCIDs

Mária Ashaber (iD) https://orcid.org/0000-0002-5586-9585
Eric A Bushong (iD) http://orcid.org/0000-0001-6195-2433
Daniel A Wagenaar (iD) https://orcid.org/0000-0002-6222-761X

### Decision letter and Author response

Decision letter https://doi.org/10.7554/eLife.61881.sa1
Author response https://doi.org/10.7554/eLife.61881.sa2

## Additional files

### Supplementary files
• Transparent reporting form

## Data availability

The easiest way to access the raw electrophysiology and voltage-dye data as well as the tracing results used in this paper is through a series of Python modules that we made available at https://github.com/wagenadl/leechem-public (copy archived at https://archive.softwareheritage.org/swh:1:rev:73eee24e387e11c259a3f3fe0bd4e469048b25e6/). Included in the package is a file called "demo.py" that demonstrates the use of the modules. Table 4 lists the available VSD trials. The aligned EM volume may be accessed through the Neuroglancer instance at https://leechem.caltech.edu or by pointing SBEMViewer to https://leechem.caltech.edu/emdata. The API is documented at https://leechem.caltech.edu/emdata/help.

The following dataset was generated:

| Author(s) | Year | Dataset title | Dataset URL | Database and Identifier |
|---|---|---|---|---|
| Ashaber MA, Tomina Y, Kassraian P, Bushong EA, Kristan WB, Ellisman MH, Wagenaar DA | 2020 | Code and data access for Ashaber et al., 2021 | https://github.com/wagenadl/leechem | Github, wagenadl/leechem |

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
