## [Decision Letter]

**Acceptance summary:**

In this Research Advances article, the authors elegantly overlay an anatomical wiring diagram onto functional maps for the generation of distinct behaviors. They do this by combining voltage -sensitive dye imaging and EM reconstruction of the circuit to reveal functional synaptic locations onto a single motor neuron in 3 different behaviors. This study will be of great significance for bridging our understanding of neural basis of behaviors at the synaptic, cellular and circuit levels.

**Decision letter after peer review:**

Thank you for submitting your article "Anatomy and activity patterns in a multifunctional motor neuron and its surrounding circuits" for consideration by *eLife*. Your article has been reviewed by three peer reviewers, and the evaluation has been overseen by a Reviewing Editor and Ronald Calabrese as the Senior Editor. The following individual involved in review of your submission has agreed to reveal their identity: Christel Genoud (Reviewer #1).

The reviewers have discussed the reviews with one another and the Reviewing Editor has drafted this decision to help you prepare a revised submission.

As the editors have judged that your manuscript is of interest, but as described below that additional analyses are required before it is published, we would like to draw your attention to changes in our revision policy that we have made in response to COVID-19 (https://elifesciences.org/articles/57162). First, because many researchers have temporarily lost access to the labs, we will give authors as much time as they need to submit revised manuscripts. We are also offering, if you choose, to post the manuscript to bioRxiv (if it is not already there) along with this decision letter and a formal designation that the manuscript is "in revision at *eLife*". Please let us know if you would like to pursue this option. (If your work is more suitable for medRxiv, you will need to post the preprint yourself, as the mechanisms for us to do so are still in development.)

In this manuscript, Ashabar et al. use a combination of voltage imaging and electron microscopy to explore the important question of how a specific motor neuron participates in distinct behaviors. The authors first perform voltage imaging of a leech ganglion to capture activity patterns of a large number of neurons during three fictive motor patterns: swimming, crawling, and local bending. Next, they perform x-ray tomography and serial block-face EM in order to reconstruct the inputs onto DE-3, a motor neuron that has different activity patterns during all three behaviors. The authors suggest that inputs correlated with distinct behaviors are widely distributed across the dendritic arbor of DE-3, but that synapses from presynaptic neurons with coherent activity cluster together more frequently than would be expected by chance. The addition of an EM dataset to the previous study on voltage imaging of the leech ganglion certainly constitutes a valuable advance.

Major Concerns:

1) The premise of this study is that mapping subcellular localization of synaptic inputs onto DE-3 may explain the multifunctionality of the DE-3 motor neuron. However, it is clear from the voltage imaging experiments that many of the premotor neurons are also active during multiple behaviors.

a) How do the authors assign neurons to distinct behaviors in Figure 8 when Figures 2-4 show that the same neurons are active across multiple behaviors?

b) If the premotor neurons are also multifunctional (i.e., active during multiple behaviors), then why would one expect there to be differences in subcellular localization on the motor neuron? Given the results from the voltage imaging data, it seems that the pre-motor network should be more important for generating distinct motor patterns, rather than integration of synaptic inputs in the DE-3 motor neuron.

2) The analysis of synapse distributions is confusing. The primary finding of the paper is that the structure of the functional clusters is different for different behaviors, as different clustering parameters gave optimal correspondence between clusters and the neural activity.

a) While useful as a statistical measure, it is not clear how the F-ratio score relates to strength of clustering or coherence of neural activity. The authors should provide additional measures that provide some indication of how much functional clustering would contribute to the activity of the motor neuron.

b) Additionally, it would be useful to see the distribution of how many neurons contribute synapses to each functional cluster. If the clusters arise from individual neurons, or a very small number of neurons, then it seems obvious that they would have a high coherence (with themselves).

c) Was an analysis of # of synapses versus known connectivity strength performed (to the extent this is known)? Or, for example, the strength of coherence with DE-3 versus number of synapses?

3) Another key analysis in the paper centers around measuring the distribution of synapses along the arbor of DE-3 (Figure 8). The authors find no differences between inhibitory/excitatory synaptic distributions nor neurons participating in different behaviors. It is unclear whether using the cell body as the reference point is a good choice for these analyses, given the unipolar nature of the motor neuron. It would be worthwhile to

a) Measure the distribution of synapses along the motor neuron arbor as a function of the distance from the primary neurite, as shunting inhibition could also occur in an arbor-specific manner, or

b) Use a point on the axon as a reference point rather than the soma, as the spike initiation zone is likely nowhere near the soma.

4) Generally an analysis of the known presynaptic connectivity onto DE-3 versus what was found in this study seems to be missing. Including a wiring diagram of the known monosynaptic presynaptic connections onto DE-3 would be extremely helpful along with an indication of which of these known connections were found in the volume and which were not (cells 27, 208, 204/205, etc…)

5) A reconstruction of a juvenile DE-3 was previously reported in Pipkin et al. How do the locations and distributions of synapses in the current volume compare to that reconstruction? At what developmental stage was the leech in this study?

6) The data quality in Pipkin et al. (e.g. the ability to resolve vesicles) seems to be higher than in this volume? Is this correct? Could data quality also be a reason for different synapse counts between the two datasets?

7) Why were 51-35 = 16 neurons not able to be matched to the VSD recording? Did they shift location during EM processing? Regarding the 35-10 = 25 neurons that were not included in the analysis because they could not be matched to known neurons: If the morphology of these 25 neurons had been completely traced, it seems at least possible they could have been matched based on morphology and previous light microscopy fills. It seems like this could have been done without too much effort given the size of leech neurons and the availability of programs to skeletonize neurons relatively rapidly. The number of synapses actually analyzed in Table 1 (~100 or so) is a small fraction of those actually identified onto DE-3 and it seems like such incompleteness could significantly impact the results in Figures 8 and 9.

8) It would be useful to see that the observed clustering on DE-3R is reproducible on DE-3^L.^ Given that the final number of neurons used in this analysis is small, is this feasible?

9) Is it possible to reconstruct the contralateral DE-3 homolog to examine contact locations with the analyzed DE-3 and whether there is an anatomical signature of an electrical synapse?

[Editors' note: further revisions were suggested prior to acceptance, as described below.]

Thank you for submitting your article "Anatomy and activity patterns in a multifunctional motor neuron and its surrounding circuits" for consideration by *eLife*. Your article has been reviewed by three peer reviewers, and the evaluation has been overseen by a Reviewing Editor and Ronald Calabrese as the Senior Editor. The following individual involved in review of your submission has agreed to reveal their identity: Christel Genoud (Reviewer #1).

The manuscript is greatly improved after the first round of revisions. However, reviewers point out several places where clarifications and/or revising claims is essential. Please address these points.

Reviewer #1:

All the requests raised by the first review have been carefully considered. The new figures have positively improved the clarity of the results. The added analysis have strengthened the results and conclusions.

Reviewer #2:

The revisions made some of the analyses and conclusions more clear. We have four remaining concerns, intended to help explain and clarify the approach for a general scientific audience.

1) The revised Materials and methods section clarifies the process used to classify neurons as belonging to distinct behaviors. This does not, however, justify that it is appropriate to classify neurons in this manner. It appears as if most neurons that have been classified as participating in one behavior exhibit similar levels of activity during other behaviors. This is also clear from Figure 4J in the previous (original) paper. The authors should provide some justification for assigning neurons to specific behaviors. Wouldn't the tree analysis still assign neurons to single behaviors even if all neurons participated in all behaviors? Would the assignment be repeatable across individual animals?

2) In response to our previous comment about premotor neurons being multifunctional (ie, participating in multiple behaviors), the authors added a statement that "DE-3 may be a relatively passive integrator of its inputs." I find this confusing. The synapse clustering analysis seems to suggest that DE-3 is not a passive integrator of inputs. If it were, would one not expect there to be no clustering at all?

3) In response to our concern about the interpretability of the F-Ratio, the authors provide some additional description in the Results and Materials and methods. This is helpful, but it would still be useful to provide some context for readers unfamiliar with this ratio. This context could be provided with some examples of F-ratio values for highly clustered or randomly distributed synapses.

4) It would be helpful to compare the coherence analyses in this manuscript with that from the previous (original) paper. They do not always appear similar across the two manuscripts. This should be discussed, especially given the data quality concerns between this paper and the previous one.

Reviewer #3:

The authors addressed most of the issues pointed out in the initial review adequately. There are a few additional points I had:

1) I still cannot see the dataset at: https://leechem.caltech.edu/. It leads to a 502 Gateway Error.

2) I think the correlation analysis of synapse number vs coherence should be added as a supplement to the paper, despite it yielding a statistically insignificant correlation.

3) This sentence in the Discussion should be corrected: “We reconstructed its entire arborization and traced all of its presynaptic partners back to their somata.” This is not true based on the number of untraceable synapses, right? The Abstract contains a similar claim.

4) Including details of what limited the data quality would help future experimenters decide on imaging parameters. Perhaps adding a “Comment on data quality” to the Materials and methods section?

---

## [Author Response]

Major concerns1) The premise of this study is that mapping subcellular localization of synaptic inputs onto DE-3 may explain the multifunctionality of the DE-3 motor neuron. However, it is clear from the voltage imaging experiments that many of the premotor neurons are also active during multiple behaviors.a) How do the authors assign neurons to distinct behaviors in Figure 8 when Figures 2-4 show that the same neurons are active across multiple behaviors?

While it is true that many neurons are active across multiple behaviors, their degree of activity is not uniform across behaviors. We have added a Materials and methods subsection “Tree analysis” to explain the assignment.

b) If the premotor neurons are also multifunctional (i.e., active during multiple behaviors), then why would one expect there to be differences in subcellular localization on the motor neuron? Given the results from the voltage imaging data, it seems that the pre-motor network should be more important for generating distinct motor patterns, rather than integration of synaptic inputs in the DE-3 motor neuron.

We added a sentence to the Discussion to address this point: “Since most presynaptic partners are multifunctional, the pre-motor network undoubtedly plays an important role in generating distinct motor patterns. Our finding suggests that DE-3 may be a relatively passive integrator of its inputs.”

2) The analysis of synapse distributions is confusing. The primary finding of the paper is that the structure of the functional clusters is different for different behaviors, as different clustering parameters gave optimal correspondence between clusters and the neural activity.a) While useful as a statistical measure, it is not clear how the F-ratio score relates to strength of clustering or coherence of neural activity.

The F-ratio does not directly relate to the strength of clustering, or to the coherence. Rather, the F-ratio captures the degree to which neurons within a spatial cluster also grouped together in the coherence plot for a given behavior. In other words, it quantifies the link between spatial clustering and functional relatedness. We have added text in Results and Materials and methods to clarify this, including a new Box that outlines the procedure.

The authors should provide additional measures that provide some indication of how much functional clustering would contribute to the activity of the motor neuron.

The role of synaptic clusters in vivo or ex vivo is a topic of ongoing research. In particular, how the background activity of in vivo conditions affects integrative dendritic properties is not yet fully understood. To answer the reviewer’s request would require a detailed biophysical model, which we deem out of the scope of the current work. Such a model *was* constructed in a recent study of mouse CA1 pyramidal cells, in which the authors benchmarked characteristics of Na^+^ spikes, the amplitude distribution of synaptic potentials, the integration of NMDA receptors, voltage thresholds, and the functional role of K^+^ channels as predicted by their model against the relevant values derived from in vitro conditions (Ujfalussy and Makara, 2020).

Our aim was rather to address another fundamental aspect underlying the functional role of synaptic clusters: If individual dendritic branches are to serve as single computational compartments, connectivity has to be not only specific between neurons, but precise with respect to individual dendritic branches. This precision could arise for instance through spontaneous neural activity or NMDA receptor signaling, such that synaptic clusters displaying synchronous activity are established through a branch-specific “fire-together-wire-together” rule (Kleindienst et al., 2011).

To assess this level of connection precision, it is fundamental to differentiate between clusters of synapses from the same presynaptic neuron vs. those from different presynaptic neurons, a feat which requires the precision of Electron Microscopy, and which could not been achieved in previous studies of synaptic clustering based on light microscopy (Druckmann et al., 2014).

Our finding that spatially clustered synapses display synchronized activity confirms observations on synchronous proximal synapses from previous studies (Kleindienst et al., 2011, Takahashi et al., 2012). Our analysis also serves as a “proof of concept”, illustrating what can be achieved by combining functional recording and EM reconstruction, which we hope will become standard in future connectomics studies.

We have added additional text in the Discussion section that seeks to convey the above ideas in a slightly more concise form.

b) Additionally, it would be useful to see the distribution of how many neurons contribute synapses to each functional cluster. If the clusters arise from individual neurons, or a very small number of neurons, then it seems obvious that they would have a high coherence (with themselves).

The F-ratio, like the F-statistic used in ANOVA, is actually immune to the size of the clusters.

c) Was an analysis of # of synapses versus known connectivity strength performed (to the extent this is known)? Or, for example, the strength of coherence with DE-3 versus number of synapses?

We have extensively analyzed the correlation between the number of synapses and the magnitude of the coherence of presynaptic partners with DE3-R during behaviors. However, even though almost all correlation coefficients were positive, none of them were *significantly* positive.

In particular, we calculated the Pearson as well as the rank-based Spearman correlation coefficients for swimming, crawling, and local bending. We found no statistically significant relationship between the number of synapses and the magnitude of the coherence of a presynaptic partner with DE3-R for any of the trials (all *p* > 0.05). As an example, we present here the results for swim trial 1 in Author response image 1. Results for all trials are summarized in Author response table 1.

**Author response image 1. sa2fig1:** 

**Author response table 1. resptable1:** Correlation between synapse count and coherence magnitude.

Trial	Pearson	*p*-value	Spearman	*p*-value
Swim 1	0.190	0.342	0.057	0.78
Swim 2	0.139	0.489	0.113	0.576
Crawl 1	0.173	0.387	-0.045	0.823
Crawl 2	0.154	0.443	0.139	0.49
Local Bend 1	0.152	0.450	-0.014	0.944
Local Bend 2	0.202	0.311	0.117	0.562
Local Bend 3	-0.279	0.158	-0.193	0.336
Local Bend 4	0.072	0.721	-0.047	0.816

In a second analysis, we weighted each synapse inversely proportional to its distance to the soma of DE-3^R^. Again, none of the trials resulted in statistically significant correlation coefficients. As an example, we again present a visualization for swim trial 1 and a summary for all trials in Author response image 2 and Author response table 2.

**Author response table 2. resptable2:** Correlation between weighted synapse count and coherence magnitude.

Trial	Pearson	p-value	Spearman	p-value
Swim 1	0.251	0.206	0.160	0.425
Swim 2	0.158	0.431	0.118	0.558
Crawl 1	0.134	0.504	-0.153	0.447
Crawl 2	0.067	0.740	-0.068	0.737
Local Bend 1	0.149	0.459	-0.028	0.891
Local Bend 2	0.268	0.177	0.194	0.333
Local Bend 3	-0.240	0.227	-0.089	0.661
Local Bend 4	0.113	0.575	0.094	0.641

In conclusion, no consistent relationship between synapse count and the strength of functional coherence of a presynaptic neuron and DE-3^R^ was found. This somewhat surprising result could be explained by variability in the strength of individual synapses or by other factors that affect the impact of individual synapses on the activity of the postsynaptic neuron, such as non-linearities associated with synaptic clusters.

3) Another key analysis in the paper centers around measuring the distribution of synapses along the arbor of DE-3 (Figure 8). The authors find no differences between inhibitory/excitatory synaptic distributions nor neurons participating in different behaviors. It is unclear whether using the cell body as the reference point is a good choice for these analyses, given the unipolar nature of the motor neuron. It would be worthwhile toa) Measure the distribution of synapses along the motor neuron arbor as a function of the distance from the primary neurite, as shunting inhibition could also occur in an arbor-specific manner, orb) Use a point on the axon as a reference point rather than the soma, as the spike initiation zone is likely nowhere near the soma.

These are excellent suggestions. We have added a supplemental figure with histograms of distances between synapses and the trunk. These did not reveal obvious structure either, which we noted in the Discussion. We created another set of histograms with distance measured from the point where the axon enters the contralateral dorsal posterior nerve, again with no significant results. In the interest of space, we have not included these histograms in the present submission, but we would not object to adding them.

4) Generally an analysis of the known presynaptic connectivity onto DE-3 versus what was found in this study seems to be missing. Including a wiring diagram of the known monosynaptic presynaptic connections onto DE-3 would be extremely helpful along with an indication of which of these known connections were found in the volume and which were not (cells 27, 208, 204/205, etc…)

Adding a wiring diagram is an excellent idea. We created a figure that shows monosynaptic connections onto DE-3 (new Figure 10) and added a paragraph related to the diagram to the Discussion section.

5) A reconstruction of a juvenile DE-3 was previously reported in Pipkin et al. How do the locations and distributions of synapses in the current volume compare to that reconstruction? At what developmental stage was the leech in this study?

The leech used in our study was an adult specimen. We have clarified that important point in both the Results and Materials and methods sections. We have inserted a sentence that explains that, unlike Pipkin et al., we did not find output synapses from DE-3 onto other cells. At a gross level, the input synapses onto DE-3 were distributed similarly in our sample as in the juvenile.

6) The data quality in Pipkin et al. (e.g. the ability to resolve vesicles) seems to be higher than in this volume? Is this correct? Could data quality also be a reason for different synapse counts between the two datasets?

Our image quality is indeed not as high as in Pipkin et al., 2016. There were three reasons for that: (1) To be able to image a much larger volume in a reasonable timespan, we had to accept a shorter per-pixel dwell time. (2) To reduce charging artifacts in the larger volume, we used gas injection (Deerinck et al., 2018) which had a small negative impact on signal to noise. (3) During our acquisition run, the microscope suffered from a hardware defect involving a bad cable. This last issue probably accounted for the majority of the difference. It is certainly possible that image quality had an effect on synapse counts, although it should be noted that our synapse count is right in between the counts Pipkin et al. report for the two DE-3 cells in their dataset.

7) Why were 51-35 = 16 neurons not able to be matched to the VSD recording? Did they shift location during EM processing? Regarding the 35-10 = 25 neurons that were not included in the analysis because they could not be matched to known neurons: If the morphology of these 25 neurons had been completely traced, it seems at least possible they could have been matched based on morphology and previous light microscopy fills. It seems like this could have been done without too much effort given the size of leech neurons and the availability of programs to skeletonize neurons relatively rapidly. The number of synapses actually analyzed in Table 1 (~100 or so) is a small fraction of those actually identified onto DE-3 and it seems like such incompleteness could significantly impact the results in Figures 8 and 9.

The neurons that could not be matched between EM and VSD did indeed shift during sample preparation. It is true that partial matching could have been done based on morphology. Unfortunately, given our image quality, complete tracing of all the neurons would have taken several years of effort. It is absolutely true that the incompleteness could have an impact on results. We are presently preparing for a second EM run, which should yield much higher image quality for future analysis.

8) It would be useful to see that the observed clustering on DE-3R is reproducible on DE-3^L.^ Given that the final number of neurons used in this analysis is small, is this feasible?

It would be very interesting to trace DE-3^L^. Unfortunately, we lack the resources to do so, as tracing DE-3^R^ with its presynaptic partners took almost two years. Tracing DE-3^L^ would likely take another year of work.

9) Is it possible to reconstruct the contralateral DE-3 homolog to examine contact locations with the analyzed DE-3 and whether there is an anatomical signature of an electrical synapse?

We partially traced DE-3^L^, but only insofar as needed to connect synapses to the soma. Our tracing includes three areas where the two neurons meet each other. Simply by inspecting relevant sections of the EM data, we could not confidently confirm any signature of electrical synapses (Fan et al., 2005). It is possible that future more sophisticated image analysis will change this picture.

[Editors' note: further revisions were suggested prior to acceptance, as described below.]

Reviewer #2:The revisions made some of the analyses and conclusions more clear. We have four remaining concerns, intended to help explain and clarify the approach for a general scientific audience.1) The revised Materials and methods section clarifies the process used to classify neurons as belonging to distinct behaviors. This does not, however, justify that it is appropriate to classify neurons in this manner. It appears as if most neurons that have been classified as participating in one behavior exhibit similar levels of activity during other behaviors. This is also clear from Figure 4J in the previous (original) paper. The authors should provide some justification for assigning neurons to specific behaviors. Wouldn't the tree analysis still assign neurons to single behaviors even if all neurons participated in all behaviors? Would the assignment be repeatable across individual animals?

We added a sentence to the end of the “Tree analysis” Materials and methods subsection to address this concern.

2) In response to our previous comment about premotor neurons being multifunctional (ie, participating in multiple behaviors), the authors added a statement that "DE-3 may be a relatively passive integrator of its inputs." I find this confusing. The synapse clustering analysis seems to suggest that DE-3 is not a passive integrator of inputs. If it were, would one not expect there to be no clustering at all?

We agree that the sentence was confusing and removed it.

3) In response to our concern about the interpretability of the F-Ratio, the authors provide some additional description in the Results and Materials and methods. This is helpful, but it would still be useful to provide some context for readers unfamiliar with this ratio. This context could be provided with some examples of F-ratio values for highly clustered or randomly distributed synapses.

This is a great suggestion. We added a new figure supplement (Figure 11—figure supplement 3) to implement it, along with additional explanation in the Materials and methods section that now includes an explicit equation.

4) It would be helpful to compare the coherence analyses in this manuscript with that from the previous (original) paper. They do not always appear similar across the two manuscripts. This should be discussed, especially given the data quality concerns between this paper and the previous one.

We added a sentence to emphasize the considerable variation that always exists between animals in the coherence values of individual cells. It may also be worth noting that the coherence plots in the original paper were referenced to cells DI-1 (for swimming), AE (for crawling), or the stimulus (for local bending), rather than to DE-3 as in this paper.

Reviewer #3:The authors addressed most of the issues pointed out in the initial review adequately. There are a few additional points I had:1) I still cannot see the dataset at: https://leechem.caltech.edu/. It leads to a 502 Gateway Error.

Apologies. When the server rebooted, the software did not automatically restart. We have now fixed this shortcoming.

2) I think the correlation analysis of synapse number vs coherence should be added as a supplement to the paper, despite it yielding a statistically insignificant correlation.

We particularly appreciate this recommendation. Looking at the data again, we realized we had never tested whether on average across trials the correlation coefficients were statistically significant. We have now performed that test and found that, in fact, they were. These results are presented in the new Figure 9 and corresponding text in Results and in Materials and methods.

3) This sentence in the Discussion should be corrected: “We reconstructed its entire arborization and traced all of its presynaptic partners back to their somata.” This is not true based on the number of untraceable synapses, right? The Abstract contains a similar claim.

We removed the words “entire” and “all,” both here and in the Abstract and added a qualifying phrase.

4) Including details of what limited the data quality would help future experimenters decide on imaging parameters. Perhaps adding a “Comment on data quality” to the Materials and methods section?

We have added a figure (Figure 6—figure supplement 2) that estimates true image resolution based on power spectra of image pixel intensities and added text to the Materials and methods section to describe our findings.